# Language Without Borders: A Dataset and Benchmark for Code-Switching Lip Reading

Xueyi Zhang[1,2,4,*],   Chengwei Zhang[5,*],   Mingrui Lao[1,†]   Peng Zhao[1,2,4]

Jun Tang[1],   Yanming Guo[1],   Siqi Cai[2,3,4],   Xianghu Yue[3],   Haizhou Li[2,3,4],

[1] National University of Defense Technology, China
[2] The Chinese University of Hong Kong, Shenzhen, China
[3] National University of Singapore, Singapore
[4] Shenzhen Research Institute of Big Data, Shenzhen, China
[5] University of Chinese Academy of Sciences Beijing, China

## Abstract

Lip reading aims at transforming the videos of continuous lip movement into textual contents, and has achieved significant progress over the past decade. It serves as a critical yet practical assistance for speech-impaired individuals, with more practicability than speech recognition in noisy environments. With the increasing interpersonal communications in social media owing to globalization, the existing monolingual datasets for lip reading may not be sufficient to meet the exponential proliferation of bilingual and even multilingual users. However, to our best knowledge, research on code-switching is only explored in speech recognition, while the attempts in lip reading are seriously neglected. To bridge this gap, we have collected a bilingual code-switching lip reading benchmark composed of Chinese and English, dubbed CSLR. As the pioneering work, we recruited 62 speakers with proficient foundations in both spoken Chinese and English to express sentences containing both involved languages. Through rigorous criteria in data selection, CSLR benchmark has accumulated 85,560 video samples with a resolution of 1080x1920, totaling over 71.3 hours of high-quality code-switching lip movement data. To systematically evaluate the technical challenges in CSLR, we implement commonly-used lip reading backbones, as well as competitive solutions in code-switching speech for benchmark testing. Experiments show CSLR to be a challenging and under-explored lip reading task. We hope our proposed benchmark will extend the applicability of code-switching lip reading, and further contribute to the communities of cross-lingual communication and collaboration. Our dataset and benchmark are accessible at GitHub.

## 1   Introduction

Speech serves as a pivotal medium for both human communications and human-computer interactions. Lip reading, known as Visual Speech Recognition (VSR), focuses on deciphering the semantic information conveyed by a speaker through the analysis of lip movements. It holds substantial significance across diverse real-world applications, such as keyword spotting [1, 2], biometric validation [3, 4], and speech enhancement [5, 6], audio-visual synchronization [7], transforming visual speech into audible speech [8, 9], speaker recognition and verification [10–13], medical communicative aids [14, 15], and human-computer interfaces [16]. Mainstream research efforts on

---

*These authors contributed equally to this work.
†Corresponding author: Mingrui Lao (laomingrui@vip.sina.cn)

38th Conference on Neural Information Processing Systems (NeurIPS 2024) Track on Datasets and Benchmarks.

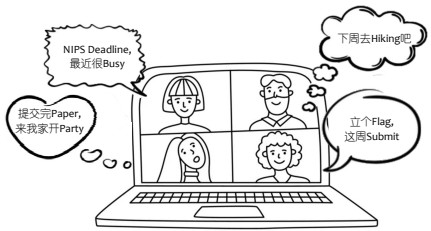

Figure 1: Multi-participant code-switching scenario in a meeting context.

sentence-level lip reading primarily focus on improving the recognition performance of monolingual content, mainly due to the lack of cross-linguistic lip reading datasets.

In the era of globalization, there is a growing interconnection among cultures and societies. This phenomenon has led to an increase in the frequency of language switching during daily conversations. As shown in Figure 1, code-switching refers to the phenomenon where individuals alternate between languages within a single conversation. This linguistic behavior may occur for various reasons. The one is the relative ease with which certain concepts can be articulated in one language as opposed to another. The other is the presence of specific technical terminology that may be more readily understood by the interlocutors when expressed in a particular language.

In this paper, we introduce the first code-switching sentence-level lip reading benchmark, termed Cross-Linguistic Code-Switching Lip Reading (CSLR). In contrast to monolingual datasets, CSLR not only contains samples labeled by the standalone single-language sentences (Chinese or English), but also involves Chinese-English code-switching. To be specific, we employ the front-facing camera on smartphones to record 62 speakers, thereby forming the dataset with a total of 85,560 valid and high-quality facial videos under the rigorous criteria of data selection. In addition, motivated by the code-switching speech recognition, we introduce various evaluation metrics including Word Error Rate (WER), Character Error Rate (CER), and Mixture Error Rate (MER) for benchmark testing. Built upon our proposed CSLR, we leverage the widely-utilized yet competitive lip reading models as backbones, and further transfer the state-of-the-art methods from code-switching speech recognition. Comprehensive experiments reveal the recognition difficulty of CSLR with unsatisfied accuracy, and qualitative analysis points out the recognition confusion between bilingual languages. We envision that CSLR will facilitate research on a new class of code-switching algorithms for the challenging cross-linguistic lip reading task.

## 2   Related Work

Table 1: Summary statistics for different publicly available lip reading datasets. These datasets contain only video samples of different lengths of individual languages.

| Datasets | Year | Level | Language | Speakers | Source | Hours |
|---|---|---|---|---|---|---|
| AVICAR [17] | 2004 | Letter | EN | 86 | Car | 33 |
| GLIP [18] | 2022 | Word | DE | 100 | Media | 80 |
| Persian [19] | 2022 | Word | PE | 1800 | Media | 30 |
| LRRo [20] | 2020 | Word | RU | 40 | Lab+Media | 25 |
| GRID [21] | 2006 | Letter | EN | 33 | Lab | 28 |
| OuluVS2 [22] | 2015 | Phrase | EN | 53 | Lab | 2 |
| LRW [23] | 2016 | Word | EN | 86 | Media | 111+ |
| LRS2 [24] | 2017 | Sentence | EN | 1000+ | Media | 225+ |
| CMLR [25] | 2019 | Sentence | CN | 11 | Media | 86+ |
| LRW-1000 [26] | 2020 | Word | CN | 1000+ | Media | 57+ |
| **CSLR** (this work) | **2024** | **Sentence** | **CN/EN/Code-Switching** | **62** | **Phone** | **71.3** |

### 2.1   Lip Reading Dataset

Lip reading datasets typically consist of video segments of speakers, with the texts of spoken content. It can be categorized into character-level, word-level, and sentence-level according to speech length.

Table 1 enumerates the widely-used lip reading datasets, supplying information such as the type of language, the number of speakers, and the form of data recording.

The commonly used word-level datasets, LRW [23] and LRW1000 [26], focus on modeling short-term information. In contrast, sentence-level datasets like LRS2 [27] and CMLR [25] extend models' capabilities to recognize sentences, which require fine-grained interactions among local and global contextual cues for effective long-term context modeling. Moreover, these datasets can be used individually and in combination for single and multilingual identification. *It is noteworthy that, however, it is inaccessible to directly blend datasets from different languages for a multilingual lip reading dataset. It can be explained by the fact that, the standard CSLR paradigm requires multiple languages existing in one single sample.*

### 2.2 Lip Reading Model

Different from action recognition models [28], which typically use single-stage networks, lip reading models utilize a two-stage architecture with front-end and back-end networks for feature extraction and sequence modeling, respectively. The front-end model often adopts architectures like 3D+2D Convolutions [29], with enhancements such as Squeeze-and-Excitation [30] and Temporal Shift Model [31] to improve local feature modeling. The back-end network mainly employs structures like RNN [32], TCN [29], and Transformer [33] to model long-term contextual relationships. *Although some methods attempt to use a single network for multilingual learning [34, 35], combining linguistic commonalities to a certain extent, none have achieved the CSLR task.*

### 2.3 Code-Switching

Code-switching has achieved notable success in the field of speech recognition [36], but it remains unexplored in lip reading. Publicly available and widely used code-switching speech recognition datasets, such as ASRU2019 [37], have facilitated advancements in this area. Methodologically, code-switching in speech recognition has evolved from multi-encoder architectures [38] to single-encoder hybrid expert fusion architectures [36, 39, 40]. This paper introduces the first code-switching lip reading dataset and validates classic and commonly used backbone network. Furthermore, we adapt and transfer both classical and state-of-the-art code-switching methods from speech recognition to lip reading.

## 3 Benchmark

### 3.1 Task definition

For conventional sentence-level lip reading, we start with a video sample containing the speaker's lip region of interest, denoted as $Video \in \mathbb{R}^{T \times W \times H \times 3}$, where $T$ represents the number of frames, and $W$ and $H$ imply the spatial size. We refer the corresponding one-hot labels of the contained subwords as $y$, where the subwords belong to either $Sub_{en}$ or $Sub_{cn}$. The optimization goal of our network is to learn the mapping from video to labels, which can be formulated as:

$$\hat{y} = f(Video, \theta) \tag{1a}$$

$$\theta^* = \arg \min_{\theta} \text{Loss}_{ctc}(\hat{y}, y), \tag{1b}$$

where $f(\cdot, \theta)$ is the mapping function, and $\hat{y}$, $\theta$ refer to the prediction results and the network parameters, respectively. The whole training process seeks to optimize model parameters $\theta^*$ to minimize the CTC loss between the predictions and the labels.

It is noteworthy that, unlike the conventional monolingual lip reading with less uncertainty, the subwords contained in each sample's label may not only be derived from the single language ($Sub_{en}$ or $Sub_{cn}$), but also from both of them ($Sub_{en}$ and $Sub_{cn}$). We expect the network to yield both Chinese characters and English words by modeling the lip movements, with the capacity for simultaneous adaptation to different languages.

### 3.2 Dataset

To bridge the gap of code-switching lip reading, and facilitate the related research, we construct the first bilingual lip reading dataset accessible to the public, which not only includes both monolingual

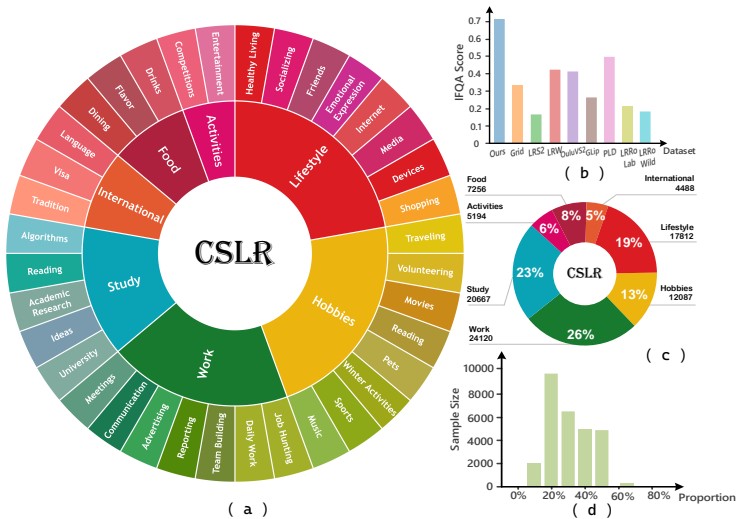

Figure 2: The dataset statistic of CSLR. (a) The abbreviated hierarchical categories structure of CSLR. (b) The fine-grained facial quality score evaluated by IFQA. (c) The quantity distribution of CSLR. (d) The sample number distribution of the proportion of Chinese characters in labels of code-switching samples in CSLR.

Chinese and English samples, but also the Chinese-English bilingual code-switching samples. To guarantee the quality of dataset collection, we enlisted a cohort of 62 volunteers who are proficient in Chinese-English bilingual oral communication. Additionally, to amplify the practical utility of the dataset, we meticulously designed a collection of 300 Chinese-English code-switching sentences derived from real-world scenarios, which is achieved by questionnaires. The involved topics in our collected dataset are summarized in Figure 2(a), while the number of samples under each topic is depicted in Figure 2(c). Note that a single video can be tagged with more than one topic. The data underwent rigorous quality control by human professionals, resulting in a total of 85,560 valid video clips. To enhance the challenge of the dataset, the proportion of independent Chinese, English, and Chinese-English code-switching samples is maintained at a 1:1:1 ratio. Our goal is for one model to not only recognize different languages but also identify the transitions between them.

The dataset contains two folders: training and testing. The training folder contains 64,336 video files, while the testing folder includes 21,224 video files. Each video is 3 seconds long, with a frame rate of 30 FPS. Files are named in an organized and straightforward way, with details like participant numbers and speech types included in the filenames. This helps users quickly identify important information about each file just by its name. All files are provided in widely compatible formats, such as .mp4, to ensure usability across various operating systems and software platforms.

### 3.3 Evaluation metrics

Considering the unique contextual attributes of our dataset, we carefully calibrate our evaluation metrics to align with the intrinsic characteristics of the involved linguistic elements. Inspired by the lexical and syntactical nuances of the Chinese and English languages respectively, we customize three metrics to comprehensively validate the efficacy of our proposed lip reading model as follows:. 1) Character Error Rate (CER) primarily evaluates the Chinese segments, through measuring the rate of character-level misdetection based on character units of Chinese. 2) Word Error Rate (WER) turns to estimate English whose basic linguistic unit is a word, by calculating the rate at which English words are correctly deciphered. 3) Mixture Error Rate (MER) serves as a holistic evaluation metric that takes into account of both English and Chinese instances within our training samples. Specifically, it evaluates characters and words as the basic units for Chinese and English, respectively.

## 4 Dataset Collection

In this section, we will detail the process of dataset collection, including the data collection tools, data quality review procedures, and data preprocessing.

### 4.1 LipGather APP

To facilitate the collection of visual-auditory-textual data in lip reading, We developed a mobile app specifically for collecting data. Its interface is depicted in the Figure 3. It **includes** comprehensive automated guidance to make data collection more convenient for participants. Details on the specific use of the app and important considerations can be found in Appendix A. Video is recorded by the phone's front camera, with the resolution set to 1920x1080 and the standard frame rate set at 30 frames per second. To prevent initial data leaks, we set up a cloud server. This approach strengthens the protection of user privacy while increasing the reliability and speed of data transfers, including quicker uploads and downloads.

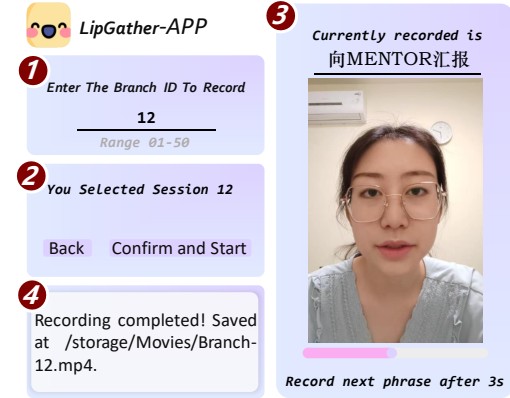

Figure 3: The interface of LipGather APP. **Step 1:** Selecting ID to record, **Step 2:** Start record. **Step 3:** Read the sentence. **Step 4:** Save and upload the video.

### 4.2 Data Quality Review

We conducted a rigorous quality check on the initially-captured 91,980 videos with assistance from 62 recruited volunteers. Specifically, they conduct a careful manual review by listening to each audio clip. This process screens out 6,420 low-quality videos, the remaining 85,560 high-quality videos for the final dataset. We use high-quality smartphone cameras for recording to ensure the dataset's excellence, as shown by the samples in Figure 4. Videos recorded with these cameras proved to be of superior visual quality than those captured with standard webcams or sourced from the internet.

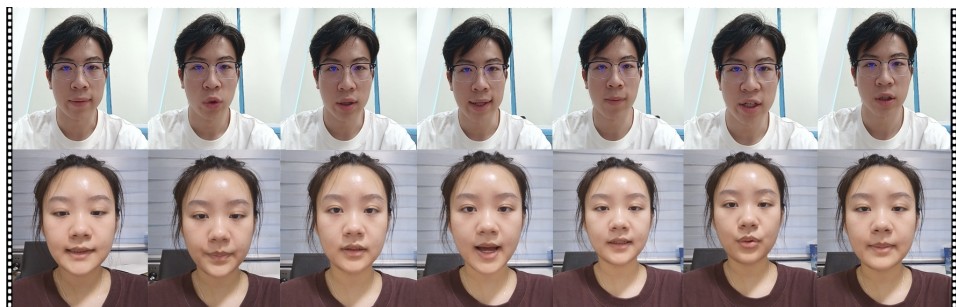

Figure 4: Examples of CSLR dataset. We took random frames from the video recorded by two speakers, a man and a woman, respectively.

To quantitatively assess the quality of our dataset, we employ an evaluation method known as Image Fidelity Quality Assessment (IFQA). This method, based on an adversarial framework, measures the image quality centered on the human face and captures subtle variations that significantly impact human visual perception and model recognition accuracy. IFQA enables us to objectively evaluate the **face quality** of critical areas within our dataset, especially the facial regions crucial for lip reading. The quantitative evaluation results of our dataset, as displayed in Figure 2(b), confirm that the data we collected surpasses other sources in terms of quality scoring.

### 4.3 Pre-processing

The recorded video often contains irrelevant background information. Research has shown that networks focus primarily on the lips and surrounding areas for lip reading [41]. Therefore, for computational efficiency and accuracy, most methods use the lip region of interest (Lip ROI) as input. Previous datasets often used the Dlib [42] library for facial landmark detection, but it showed poor accuracy and frequent detection failures, leading to unstable lip region localization and hindering the modeling of subtle lip movements. We use state-of-the-art deep learning-based methods for

face localization and lip region cropping. SPIGA [43] employs deep convolutional neural networks with facial model prior knowledge, achieving excellent localization accuracy even in challenging environments or large facial rotations. For each video, we extract frame-level facial landmarks describing the contours of the face, eyes, nose, and lips. We use the mean 2D coordinates of the 20 lip landmarks in each frame as the center of the lip region, represented as $Lip_{\text{Center}} \in \mathbb{R}^{T \times 2}$ for the entire video. To stabilize the lip positions, we apply center point filtering to account for frame-by-frame landmark detection variability and rapid lip movements. This involves averaging the lip center coordinates over the $k$ frames before and after the current frame ($k = 2$), with the mean position used as the coordinate for the current frame. The filtering process is defined as:

$$Lip^i_{\text{Center Filtered}} = \frac{1}{F} \sum_{j=i-k}^{i+k} Lip^j_{\text{Center}}, \qquad (2)$$

where $i$ denotes the current frame, and $j$ denotes the neighboring frames. This processing mitigates the impact of unstable landmark localization. We use this 2D coordinate as the center to crop a square region with a certain width as the extracted lip region. Finally, we resize all cropped lip regions to a uniform size, obtaining the final cropped lip region.

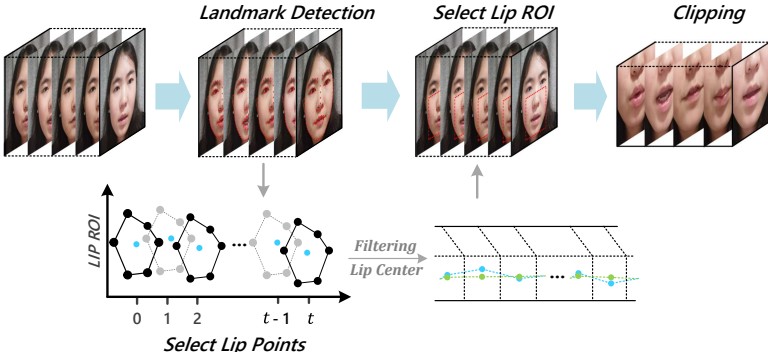

Figure 5: The preprocessing pipeline of CSLR. We first detect the facial landmarks of each frame in the video and calculate the lip center point. Then filter the spatial coordinates of the center point, and crop the lip ROI.

## 5 Experiments

The lip reading network encoder embodies two distinct networks, the front-end and back-end. The front-end extracts short-term features via a 3D CNN layer and multiple 2D CNN layers while employing global average pooling over spatial dimensions for temporal feature maps. The back-end contributes to the long-term contextual modeling and retains the temporal dimension for subword prediction at each time-step.

### 5.1 Evaluation on Backbone

The choice of front-end networks is relatively consistent across different lip reading tasks. Balancing efficiency and accuracy, we chose ShuffleNet [44] as the front-end. We explore various widely-used lip reading back-end models: BiGRU, MSTCN, and Conformer. Each model possesses its unique structural advantages: BiGRU is primarily driven by an RNN structure, MSTCN is mainly composed of a CNN layout, and Conformer is predominantly built on the Self-Attention mechanism. By examining these models, we were able to discern the potential adaptation of each network type under code-switching environments. Our examination aims to guide future research towards model selection for code-switching lip reading tasks with an emphasis on adaptability and efficiency.

We conducted training and testing on two separate monolingual datasets, as well as on both monolingual and code-switching data, to validate the consistency of different backbones' performance. The experimental results, shown in Table 2, indicate that monolingual training achieves consistent performance trends across different backbones. The MSTCN model underperforms due to its limited

Table 2: Comparison of CER, WER, and MER for monolingual and code-switching on the CSLR dataset using different back-end models. Monolingual is defined here as the unique utilization of either Chinese or English single language datasets during the process of training and examination. Code-switching refers to a joint training and testing method combining both Chinese and English monolingual data as well as bilingual alternating data.

| Front-end | Back-end | Monolingual | | Code-Switching | | |
|---|---|---|---|---|---|---|
| | | CN | EN | | | |
| | | CER↓ | WER↓ | CER↓ | WER↓ | MER↓ |
| | *MSTCN* | 34.36% | 53.69% | 36.80% | 55.27% | 43.56% |
| *ShuffleNet* | *BiGRU* | **30.25%** | 51.08% | 36.91% | 54.74% | 43.44% |
| | *Conformer* | 30.95% | **44.66%** | **35.32%** | **49.05%** | **40.35%** |

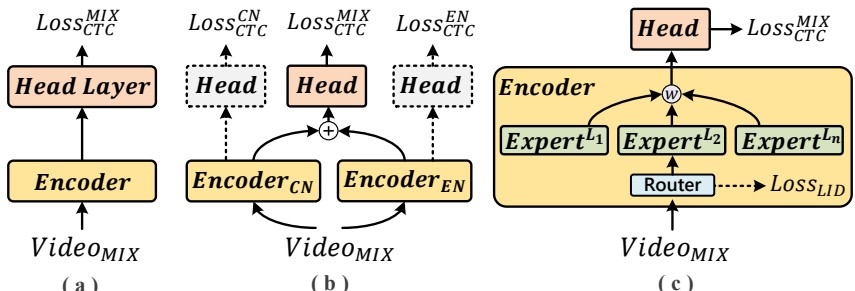

Figure 6: State-of-the-art code-switching speech recognition methods. (a) Vallina CTC. (b) Bi-Encoder CTC. The mixed loss $Loss_{CTC}^{MIX}$ can also integrate with single language mask loss $Loss_{CTC}^{EN}$ and $Loss_{CTC}^{CN}$ though Language Aware Training (LAT). (c) Language-Routing Mixture of Experts (LR-MoE).

receptive field for long sequences, and its dilation factor fails to fully address sentence-level lip reading demands. The BiGRU model ranks second in accuracy, leveraging its bidirectional structure to maintain context over time but may forget over long sequences. The Conformer model outperforms others, thanks to its attention mechanism for grasping long-range dependencies and a convolution layer that highlights local temporal features, significantly enhancing code-switching lip reading accuracy.

## 5.2 Evaluation on Code-Switching Methods

Since code-switching is not explored in lip reading, we referred to classical and state-of-the-art methods from code-switching speech recognition to test their adaptability in the lip reading code-switching task. This includes (1) Vallina CTC, (2) Bi-Encoder CTC, (3) Language Aware Training, and (4) Language-Routing Mixture of Experts (LR-MoE). The structures of the comparative methods are illustrated in Figure 6.

**Vallina CTC:** This method uses a single encoder network, $Encoder(\cdot, \theta_{\text{Mix}})$, to recognize both monolingual and code-switching samples. The encoder extracts features from the lip video $Video_{\text{Mix}} \in \mathbb{R}^{T \times H \times W \times 3}$. These features are passed through a classification head to compute prediction probabilities for each subword. The CTC loss is calculated between the predictions and labels:

$$Pred^{\text{Mix}} = \text{Head}(Encoder(Video_{\text{Mix}}, \theta_{\text{Mix}})) \tag{3a}$$

$$Loss_{\text{Mix}} = \text{CTC}(Pred^{\text{Mix}}, Label_{\text{Mix}}) \tag{3b}$$

The main drawback of this single-stream network is its inability to adapt well to code-switching scenarios due to a lack of consideration for the characteristics of different languages.

**Bi-Encoder CTC:** The bi-encoder structure enhances speech modeling by separating feature extraction for different languages. We first train two identical monolingual recognition networks on

single-language samples and load the weight. Then, we combine the extracted features through a fusion layer for unified representation learning. The model is further trained on mixed-language samples for code-switching scenarios. This process is represented as:

$$Pred_{\text{Mix}} = \text{Head}(\text{Fusion}(H^{\text{CN}} \parallel H^{\text{EN}})) \tag{4}$$

where $H^{\text{CN}}$ and $H^{\text{EN}}$ are the features extracted by the Chinese and English encoders, respectively. This bi-encoder method can partially inherit monolingual prior knowledge but may have lower fusion efficiency and lack precise language-specific guidance.

**Language Aware Training:** This method masks other languages in the label and uses an extra head for single-language supervision. The losses from the monolingual branches are combined with the mixed loss using a weighting factor $\lambda$:

$$Loss = Loss_{\text{Mix}} + \lambda(Loss_{\text{EN}} + Loss_{\text{CN}}), \tag{5}$$

where $Loss_{\text{EN}}$ and $Loss_{\text{CN}}$ are the CTC losses calculated for English and Chinese samples, respectively, with other languages masked in the labels. This approach introduces additional monolingual supervision, enhancing the learning of single-language features. However, the independent structures limit the learning of cross-language commonalities and characteristics.

**Language Adaptive Mixture-of-Experts:** This method uses Mixture of Experts (MoE) in the encoder to handle both single-language and code-switching samples without needing pre-trained monolingual encoders. A Language Identification (LiD) loss supervises the expert weights of the MoE. The process is as follows:

$$H^{\text{MoE}} = \sum_{i=0}^{L} \text{Expert}_i(H^{\text{Mix}}) \cdot W_i, \tag{6}$$

where $H^{\text{Mix}}$ are the mixed features extracted by the encoder, $\text{Expert}_i$ denotes the $i$-th expert network, $W_i$ are the fusion weights controlled by a linear layer applied to $H^{\text{Mix}}$, and $L$ is the number of experts. The LiD loss is the cross-entropy loss between $W$ and the LiD labels. This approach allows the model to adapt to different language scenarios by leveraging specific experts, enhancing language adaptability.

Table 3: Comparison of CER, WER, and MER for monolingual and code-switching recognition on the CSLR dataset using different code-switching methods.

| Methods | Code-Switching | | |
| --- | --- | --- | --- |
| | CER↓ | WER↓ | MER↓ |
| *Vallina CTC* | 35.32% | 49.05% | 40.35% |
| *Bi-Encoder* | 33.81% | 46.33% | 38.40% |
| *Bi-Encoder+LAT* | 33.54% | 46.08% | 38.13% |
| *LR-MoE CTC* | 33.01% | 45.37% | 37.54% |

We implemented code-switching methods using ShuffleNet+Conformer as the encoder, evaluating four approaches: (1) Vanilla CTC, (2) Bi-Encoder CTC, (3) Language Aware Training (an improvement on Bi-Encoder CTC), and (4) Language Adaptive Mixture-of-Expert (MoE). The results in Table 3 show that the vanilla single-stream network struggled with language switching, resulting in low accuracy. The bi-encoder model with monolingual pre-training significantly improved code-switching scenario performance by better integrating monolingual knowledge. Further incorporating a language-aware training strategy enhanced the learning of monolingual language characteristics, promoting better multilingual integration and recognition. The Language Routing MoE achieved the highest accuracy by effectively introducing linguistic prior knowledge through LiD loss on expert weights.

## 5.3 Qualitative Analysis

We conducted a statistical analysis and qualitative examination of the prediction results. Through the qualitative analysis of error cases, we found that some samples were misclassified as another language. Despite significant phonetic differences between the model's predictions and the ground truth, visual similarities in lip movements led to misrecognitions. For instance, as shown in Figure 7,

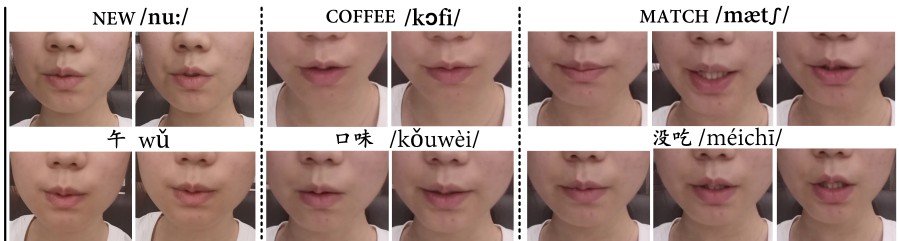

Figure 7: Visual similarities and phonetic discrepancies: an examination of misrecognitions in code-switching lip reading.

"new–/nu:/" was misrecognized as "午–/wǔ/", "没吃–/méi chī/" as "match–/mæt/", and "口味–/kǒu wèi/" as "coffee–/kɔfi/". By comparing the lip movements of speakers while speaking, we found that even with significant pronunciation differences, there are indeed substantial visual similarities in lip movements.

## 6 Limitation and Future Work

Lip reading is prone to encounter the privacy-sensitive data, as the video typically involve identifiable facial features. Current anonymization techniques are inadequate, and our dataset inevitably involves videos recorded with participants' informed consent, which cannot be thoroughly anonymized without compromising its utility.

The variation in lip movements is less numerous compared to audio, increasing the difficulty for lip reading models during the decoding process. For example, while the sounds $'p'$ and $'b'$ are distinguishable auditorily, they are visually similar and hard to differentiate. This visual ambiguity poses a remarkable challenge for code-switching, which hinders models to determine the underlying language based on lip movements. Therefore, uncovering the distinct patterns of lip movements across different languages, and decoupling the common and specific lip movement features would be a significant technical route for code-switching lip reading recognition.

## 7 Conclusion

In this paper, we introduce a novel yet practical CSLR benchmark, which contains the pioneering and large-scale Chinese and English code-switching lip reading dataset, to meet the growing practical needs of bilingual and even multilingual users. Specifically, the dataset comprises 85,560 high-quality videos captured from 62 proficient speakers in both Chinese and English, accompanied with totally over 71.3 hours of lip movement data. Based on our proposed benchmark, we leverage widely-used yet competitive lip reading models as backbones, and further conduct state-of-the-art code-switching solutions in speech recognition. Extensive experimental results demonstrate the complexity and challenges of CSLR, especially the recognition confusion between Chinese and English. We hope CSLR will facilitate and advance lip reading research, offering valuable theoretical insights into code-switching lip reading, as well as potential applications for cross-lingual communication under noisy environments.

## 8 Acknowledgments

This work is partially supported by the National Natural Science Foundation of China (Grant Nos. 62271432, 62073330), the Shenzhen Science and Technology Program (Shenzhen Key Laboratory Grant No. ZDSYS20230626091302006), the Shenzhen Science and Technology Research Fund (Fundamental Research Key Project Grant No. JCYJ20220818103001002), and the Hunan Provincial Natural Science Foundation (Grant No. 2023JJ30082). Additional support is provided by the IAF, A*STAR, SOITEC, NXP, and the National University of Singapore under the FD-fAbrICS: Joint Lab for FD-SOI Always-on Intelligent & Connected Systems project (Award I2001E0053).

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

# 9 Submissions to the NeurIPS 2024 Track on Datasets and Benchmarks

Please read the instructions below carefully and follow them faithfully.

