# Language Without Borders: A Dataset and Benchmark for Code-Switching Lip Reading Supplementary Material

This supplement to our main paper, "Language Without Borders: A Dataset and Benchmark for Code-Switching Lip Reading," includes detailed descriptions of the dataset collection methods, a comprehensive data card, and datasheets. Additionally, we provide licensing information for the dataset, along with an author statement affirming adherence to the license. Further discussions on the societal impact are included, covering cultural context and privacy considerations. Implementation details of the methods applied to the dataset are also provided.

## 1 Methodology and Data Collection Process

### 1.1 Pre-Recording Preparations

To streamline the data collection process and ensure conscientious participation, we design and develop a comprehensive mobile application specifically for this purpose. This application, illustrated in Figure 3, not only facilitates the usages of participants, but also ensures the integrity and uniformity of the collected data.

Prior to the commencement of the recording, participants are adequately briefed about the entire data collection process and all necessary precautions. This includes detailed instructions for downloading and installing our application, important pre-requisites for successful data collection such as securing a quiet environment for recordings. It guarantees that the participant's face is fully within the video frame and directly facing the camera, and avoiding the presence of additional faces in the recording frame.

It is of fundamental importance that during the recording, participants are advised to hold their phone with one hand while maintaining an optimal distance from the camera to achieve clear and properly framed video images. To avoid any distractions or impediments during the recording session, participants are recommended to disable notification alerting from various apps like WeChat or any others that could potentially obstruct the recording interface's prompts.

### 1.2 Recording Pipeline

To ensure a seamless and productive recording session, we provide participants with an advanced briefing on the exact recording pipeline. Each recording session contains sets of 30 sentences. It is tailored to simulate to stimulate regular breaks during the tasks, thereby mitigating fatigue and maintaining a positive task experience and maximum performance levels.

For each sentence, the recording duration is precisely defined as three seconds, after which the application automatically redirects to the recording interface for the next sentence. However, should errors be identified or the recording session be interrupted for any unforeseen reasons, participants are ushered back to the main menu to recommence recording.

38th Conference on Neural Information Processing Systems (NeurIPS 2024) Track on Datasets and Benchmarks.

### 1.3 Additional Instructions and Provisions

The video recordings are captured via the phone's frontal camera, with the resolution set at 1920x1080. The standard video frame rate is meticulously set at 30 frames per second to ensure optimal image clarity and timing accuracy.

In order to fortify the stability of the data collection application and alleviate the burden of data uploading, we strategically set up an encrypted cloud server. This ensures robust data transmission, after which all collected data are downloaded to a local server where they undergo further storage and processing.

## 2 Dataset Structure

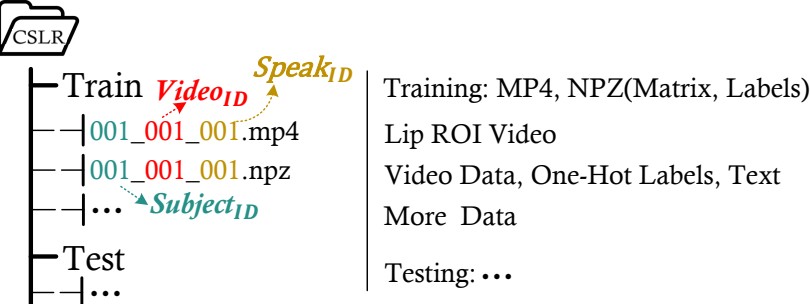

Figure 1: Dataset structure showing the organization of training and testing data, including MP4 and NPZ files. Samples are stored in NPZ format for efficient data processing and in MP4 format for easy visual inspection. Each file name follows the format $Subject_{ID}\_Video_{ID}\_Speak_{ID}$.

Table 1: Data card of the CSLR dataset. We present the first code-switching lip reading dataset with detailed category information. The number of speakers and the recording duration are sufficient for training and validating deep learning models.

| Category | Data |
|---|---|
| Size of Dataset | 71.1 GB |
| Total Number of Videos | 85560 |
| Total Number of Speakers | 62 |
| Language | CN/EN/Code-Switching |
| Source | Phone |
| Hours | 71.4 |

As depicted in Table 1, our dataset consists of 91,980 raw samples collected from 62 speakers, among which 85,560 valid samples were selected. The storage format is shown in Figure 1 and the data card with detailed statistics is shown in Table 1. The dataset is organized into two folders: training set and testing set, specialized for model training and performance evaluation. To facilitate detailed examination of each sample, important indicators are included in the file names. For files in both the training and testing sets, we use a uniform naming format: "$Subject_{ID}\_Video_{ID}\_Speak_{ID}$", where $Subject_{ID}$ identifies the speaker and range from 0 to number of speaker, $Video_{ID}$ denotes the index of the 50 videos, $Speak_{ID}$ indicates the sentence number within the video.

To facilitate the usage, we provide samples in both mp4 and npz formats. The mp4 files can be directly opened on most graphical interface operating systems, making it convenient to view the specific content of the data. The npz format data is provided to facilitate code loading and includes the following data contents:

1. **Video**: A matrix representing lip video data with the format $Video \in R^{90 \times 96 \times 96 \times 1}$, where 90 is the number of frames, 96 is the spatial dimension of the video, and 1 denotes the gray-scale video.

2. **Label**: One-hot label of subwords corresponding to the sample, with the format $R^T$, where $T$ represents the number of subwords contained in the sample's sentence.

3. **Text**: The text content of the sentence, used for calculating WER (Word Error Rate).

# 3 Societal Impact

As mentioned in Section 7 of the main paper, our cross-linguistic lip reading benchmark introduces a novel and practical research tool. This benchmark includes a pioneering large-scale cross-linguistic lip reading dataset, addressing the growing practical needs of bilingual users. The goal of CSLR is not only to advance lip reading research and provide profound theoretical insights into code-switching lip reading but also to support the development of bilingual communities on a practical basis. However, despite the broad application prospects of these tools, they may also bring some potential societal negative impacts.

## 3.1 Cultural Context

Although cross-linguistic lip reading technology can partially eliminate boundaries of language, culture, and geography, the same dialogue content can have different meanings depending on the cultural context. Therefore, misunderstandings due to linguistic, regional, and cultural differences may still occur. We advise users not to rely solely on the results of visual speech recognition but to use it as an auxiliary tool, combining it with the speaker's tone and cultural background to better understand the intended message.

## 3.2 Privacy

If lip reading technology is misused or abused, it may capture private conversations without people's knowledge or consent, leading to various privacy issues. To address this, we proactively take necessary precautions to ensure that our data is only used for academic research. Users are required to fill out an application form, indicating whether they request full-face data or only the cropped lip ROI region. The application form must be sent from an institutional email address and include the applicant's signature. We have designated personnel responsible for processing applications, and for academic purposes, we promptly approve requests and provide the data.

# 4 Access to Dataset and Benchmark

The CSLR dataset, which is available on Google Drive as a general-purpose open repository, is the fruit of diligent collection, updates, and maintenance from the team members from the Big Speech Data Laboratory of The Chinese University of Hong Kong (Shenzhen). Users keen on accessing the dataset can complete an application form via `https://forms.gle/Ttjvxdh5mmyaxF6a7`, upon which they'll be instantly and automatically provided with a download link. In addition, instructions for the dataset's creation and experiments' execution can be visited at `https://github.com/cslr-lipreading/CSLR`.

# 5 Licence

We grant permission to use all the data we release under the CC-BY-4.0 licence. Detailed access procedures for the data and preprocessing scripts are furnished in our GitHub repository. Furthermore, we've offered a means for the dataset to be downloaded. Please be reminded that this dataset is reserved solely for research applications.

# 6 Implementation Details

In our experiments, we used the same hyperparameters for network training. We primarily referenced the PyTorch implementation of Conformer [1] for both training and testing. The baseline comparison method was derived from the implementation of MSTCN [2]. Due to the lack of open-source code

for the comparison method, we replicated it based on the descriptions provided in the corresponding paper [3–5].

To ensure that all models could converge, we trained them for 200 epochs, considering the balancing of training duration and convergence. We set the batch size to 24. The optimizer is Adam [6] with betas set to 0.9 and 0.999. We experimented with learning rates ranging from 1e-5 to 1e-2, increasing tenfold, and found that a learning rate of 1e-3 provided the best convergence. To further stabilize convergence, we employed a cosine learning rate scheduler that smoothly decreased the learning rate over epochs, and we used a warmup strategy for the first 5 epochs, linearly increasing the learning rate from 0 to 1e-3.

## 7 Datasheet for Dataset

We include a datasheet based on the framework set forward by Gebru et al. [7].

### 7.1 Motivation

- **For what purpose was the dataset created?** Was there a specific task in mind? Was there a specific gap that needed to be filled? Please provide a description.

  **A:** The dataset was created to train and evaluate code-switching lip reading methods. We implemented several state-of-the-art code-switching methods in audio speech recognition, but there is still significant room for improvement.

- **Who created the dataset (e.g., which team, research group) and on behalf of which entity (e.g., company, institution, organization)?**

  **A:** The dataset was created by the authors.

- **Who funded the creation of the dataset?** If there is an associated grant, please provide the name of the grantor and the grant name and number.

  **A:** The research is supported by National Natural Science Foundation of China (Grant No. 62271432); Internal Project of Shenzhen Research Institute of Big Data (Grant No. T00120220002); Shenzhen Science and Technology Program ZDSYS20230626091302006; and Shenzhen Science and Technology Research Fund (Fundamental Research Key Project Grant No. JCYJ20220818103001002).

- **Any other comments?**

  **A:** No.

### 7.2 Composition

- **What do the instances that comprise the dataset represent (e.g., documents, photos, people, countries)?** Are there multiple types of instances (e.g., movies, users, and ratings; people and interactions between them; nodes and edges)? Please provide a description.

  **A:** Each sample in the dataset includes a cropped lip video stored in MP4 format for easy viewing, and corresponding data in NPZ format for efficient loading. The NPZ files contain one-hot labels and the text, which are used for loss computation and performance evaluation.

- **How many instances are there in total (of each type, if appropriate)?**

  **A:** The final instance number is 85560.

- **Does the dataset contain all possible instances or is it a sample (not necessarily random) of instances from a larger set?** If the dataset is a sample, then what is the larger set? Is the sample representative of the larger set (e.g., geographic coverage)? If so, please describe how this representativeness was validated/verified. If it is not representative of the larger set, please describe why not (e.g., to cover a more diverse range of instances, because instances were withheld or unavailable).

  **A:** Our dataset is collected from 62 subjects. Given the number of speakers and the duration of recordings, it covers a wide range of lip features and speaking styles, thus providing a certain degree of representativeness.

- **What data does each instance consist of?** "Raw" data (e.g., unprocessed text or images)or features? In either case, please provide a description.

**A:** Each instance includes preprocessed lip ROI videos, along with subword labels and the text of the speech.

- **Is there a label or target associated with each instance?** If so, please provide a description.

  **A:** Each instance is labeled with specific subword one-hot labels and the text.

- **Is any information missing from individual instances?** If so, please provide a description, explaining why this information is missing (e.g., because it was unavailable). This does not include intentionally removed information, but might include, e.g., redacted text.

  **A:** No.

- **Are relationships between individual instances made explicit (e.g., users' movie ratings, social network links)?** If so, please describe how these relationships are made explicit.

  **A:** The relationships between individual instances are clearly defined using subject ID, video ID, and speak ID.

- **Are there recommended data splits (e.g., training, development/validation, testing)?** If so, please provide a description of these splits, explaining the rationale behind them.

  **A:** Yes.

- **Are there any errors, sources of noise, or redundancies in the dataset?** If so, please provide a description.

  **A:** Our dataset has been manually curated to significantly reduce errors, and efforts were made to eliminate noise sources and redundancies.

- **Is the dataset self-contained, or does it link to or otherwise rely on external resources (e.g., websites, tweets, other datasets)?** If it links to or relies on external resources, a) are there guarantees that they will exist, and remain constant, over time; b) are there official archival versions of the complete dataset (i.e., including the external resources as they existed at the time the dataset was created); c) are there any restrictions (e.g., licenses, fees) associated with any of the external resources that might apply to a dataset consumer? Please provide descriptions of all external resources and any restrictions associated with them, as well as links or other access points, as appropriate.

  **A:** It is self-contained.

- **Does the dataset contain data that might be considered confidential (e.g., data that is protected by legal privilege or by doctor–patient confidentiality, data that includes the content of individuals' nonpublic communications)?** If so, please provide a description.

  **A:** No.

- **Does the dataset contain data that, if viewed directly, might be offensive, insulting, threatening, or might otherwise cause anxiety?** If so, please describe why.

  **A:** No.

- **Does the dataset identify any subpopulations (e.g., by age, gender)?** If so, please describe how these subpopulations are identified and provide a description of their respective distributions within the dataset.

  **A:** No.

- **Is it possible to identify individuals (i.e., one or more natural persons), either directly or indirectly (i.e., in combination with other data) from the dataset?** If so, please describe how.

  **A:** The provided data consists of cropped lip ROI, with most facial features outside the ROI. Therefore, it is nearly impossible to identify individuals from the dataset.

- **Does the dataset contain data that might be considered sensitive in any way (e.g., data that reveals race or ethnic origins, sexual orientations, religious beliefs, political opinions or union memberships, or locations; financial or health data; biometric or genetic data; forms of government identification, such as social security numbers; criminal history)?** If so, please provide a description.

  **A:** No.

- **Any other comments?**

  **A:** No.

### 7.3 Collection Process

- **How was the data associated with each instance acquired?** Was the data directly observable (e.g., raw text, movie ratings), reported by subjects (e.g., survey responses), or indirectly inferred/derived from other data (e.g., part-of-speech tags, model-based guesses for age or language)? If the data was reported by subjects or indirectly inferred/derived from other data, was the data validated/verified? If so, please describe how.

  **A:** We collected the data by recording with smartphone cameras, which includes raw lip videos and speech data. To validate the suitability of our dataset, we applied various speech recognition methods to it. While these methods demonstrated the feasibility of cs recognition, there is still room for improvement in accuracy. In our view, this validates the suitability of our dataset.

- **What mechanisms or procedures were used to collect the data (e.g., hardware apparatuses or sensors, manual human curation, software programs, software APIs)?** How were these mechanisms or procedures validated?

  **A:** We used a collection app to activate the front camera for data recording and employed automated tools for data preprocessing. Additionally, expert curation was performed to remove erroneous data. Our entire automated workflow is described and deposited in the linked GitHub repository, allowing any expert to validate it.

- **If the dataset is a sample from a larger set, what was the sampling strategy (e.g., deterministic, probabilistic with specific sampling probabilities)?**

  **A:** Our dataset was self-collected without using a specific sampling strategy. The data was recorded using front cameras and collected from 62 subjects, aiming to capture a variety of speaking styles and lip features.

- **Who was involved in the data collection process (e.g., students, crowdworkers, contractors) and how were they compensated (e.g., how much were crowdworkers paid)?**

  **A:** Our subjects are students from some universities in China, and each person is paid $100 ¥$ /h.

- **Over what timeframe was the data collected? Does this timeframe match the creation timeframe of the data associated with the instances (e.g., recent crawl of old news articles)?** If not, please describe the timeframe in which the data associated with the instances was created.

  **A:** All data is recorded in 2023.

- **Were any ethical review processes conducted (e.g., by an institutional review board)?** If so, please provide a description of these review processes, including the outcomes, as well as a link or other access point to any supporting documentation.

  **A:** We have discussed the ethical problems in section 3.

- **Did you collect the data from the individuals in question directly, or obtain it via third parties or other sources (e.g., websites)?**

  **A:** From the individuals.

- **Were the individuals in question notified about the data collection?** If so, please describe (or show with screenshots or other information) how notice was provided, and provide a link or other access point to, or otherwise reproduce, the exact language of the notification itself.

  **A:** We signed an agreement with the subjects and provided the APP and recording details.

- **Did the individuals in question consent to the collection and use of their data?** If so, please describe (or show with screenshots or other information) how consent was requested and provided, and provide a link or other access point to, or otherwise reproduce, the exact language to which the individuals consented.

  **A:** They agreed to collection and use of the data in the agreement.

- **If consent was obtained, were the consenting individuals provided with a mechanism to revoke their consent in the future or for certain uses?** If so, please provide a description, as well as a link or other access point to the mechanism (if appropriate).

  **A:** If the subject wishes to withdraw, please contact us directly by email.

- **Has an analysis of the potential impact of the dataset and its use on data subjects (e.g., a data protection impact analysis) been conducted?** If so, please provide a description of this analysis, including the outcomes, as well as a link or other access point to any supporting documentation.

  **A:** No.

- **Any other comments?**

  **A:** No.

### 7.4 Preprocessing/cleaning/labeling

- **Was any preprocessing/cleaning/labeling of the data done (e.g., discretization or bucketing, tokenization, part-of-speech tagging, SIFT feature extraction, removal of instances, processing of missing values)?** If so, please provide a description. If not, you may skip the remaining questions in this section.

  **A:** We located the face and cropped the lip region from videos that included the entire face and background. Data cleaning was performed by experts who verified the consistency between the sample content and the corresponding text labels; any inconsistent samples were removed.

- **Was the "raw" data saved in addition to the preprocessed/cleaned/labeled data (e.g., to support unanticipated future uses)?** If so, please provide a link or other access point to the "raw" data.

  **A:** The original data, which includes facial information, has been preserved. If needed, it can be accessed by contacting us directly.

- **Is the software that was used to preprocess/clean/label the data available?** If so, please provide a link or other access point.

  **A:** Our preprocessing scripts are available on GitHub.

- **Any other comments?**

  **A:** No.

### 7.5 Uses

- **Has the dataset been used for any tasks already?** If so, please provide a description.

  **A:** The dataset is being introduced for the first time and has only been used for code-switching lip reading tasks. We used the dataset for selected benchmark methods as described in our paper.

- **Is there a repository that links to any or all papers or systems that use the dataset?** If so, please provide a link or other access point.

  **A:** Currently, there are no repositories linking to papers or systems that use the dataset. However, we plan to upload our dataset and code links on Paperwithcode in the near future.

- **What (other) tasks could the dataset be used for?**

  **A:** The dataset can also be used for high-quality talking face generation tasks.

- **Is there anything about the composition of the dataset or the way it was collected and preprocessed/cleaned/labeled that might impact future uses?** For example, is there anything that a dataset consumer might need to know to avoid uses that could result in unfair treatment of individuals or groups (e.g., stereotyping, quality of service issues) or other risks or harms (e.g., legal risks, financial harms)? If so, please provide a description. Is there anything a dataset consumer could do to mitigate these risks or harms?

  **A:** The collected data does not represent the general population.

- **Are there tasks for which the dataset should not be used?** If so, please provide a description.

  **A:** This dataset is to be used for research purposes only. It is not intended for clinical usage.

- **Any other comments?**

  **A:** No.

## 7.6 Distribution

- **Will the dataset be distributed to third parties outside of the entity (e.g., company, institution, organization) on behalf of which the dataset was created?** If so, please provide a description.

  **A:** The dataset will be released to the general public but not to any specific third party.

- **How will the dataset be distributed (e.g., tarball on website, API, GitHub)?** Does the dataset have a digital object identifier (DOI)?

  **A:** The dataset will be distributed as split compressed archives uploaded to Google Cloud. The processing and decompression methods are documented on GitHub.

- **When will the dataset be distributed?**

  **A:** The dataset is available after filling the application form.

- **Will the dataset be distributed under a copyright or other intellectual property (IP) license, and/or under applicable terms of use (ToU)?** If so, please describe this license and/or ToU, and provide a link or other access point to, or otherwise reproduce, any relevant licensing terms or ToU, as well as any fees associated with these restrictions.

  **A:** The dataset is distributed under CC-BY-4.0 licence.

- **Have any third parties imposed IP-based or other restrictions on the data associated with the instances?** If so, please describe these restrictions, and provide a link or other access point to, or otherwise reproduce, any relevant licensing terms, as well as any fees associated with these restrictions.

  **A:** No.

- **Do any export controls or other regulatory restrictions apply to the dataset or to individual instances?** If so, please describe these restrictions, and provide a link or other access point to, or otherwise reproduce, any supporting documentation.

  **A:** No.

- **Any other comments?**

  **A:** No.

## 7.7 Maintenance

- **Who will be supporting/hosting/maintaining the dataset?**

  **A:** Our research group at CUHKSZ will continue to host and maintain the dataset.

- **How can the owner/curator/manager of the dataset be contacted (e.g., email address)?**

  **A:** Through Email to zhangxueyi@cuhk.edu.cn or in GitHub.

- **Is there an erratum? If so, please provide a link or other access point.**

  **A:** We plan to document possible corrections to the dataset via GitHub.

- **Will the dataset be updated (e.g., to correct labeling errors, add new instances, delete instances)?** If so, please describe how often, by whom, and how updates will be communicated to dataset consumers (e.g., mailing list, GitHub)?

  **A:** We do not plan to regularly update the dataset. However, if it should be necessary, we will communicate this via GitHub.

- **If the dataset relates to people, are there applicable limits on the retention of the data associated with the instances (e.g., were the individuals in question told that their data would be retained for a fixed period of time and then deleted)?** If so, please describe these limits and explain how they will be enforced.

  **A:** No.

- **Will older versions of the dataset continue to be supported/hosted/maintained?** If so, please describe how. If not, please describe how its obsolescence will be communicated to dataset consumers.

  **A:** We do not plan to change the general structure of the dataset even with a possible update, so there should be no need for user customization in this case. In case we do change the general structure of the dataset, we will provide tools to migrate from the outdated version to the current version. We will document all updates and changes on GitHub.

- **If others want to extend/augment/build on/contribute to the dataset, is there a mechanism for them to do so?** If so, please provide a description. Will these contributions be validated/verified? If so, please describe how. If not, why not? Is there a process for communicating/distributing these contributions to dataset consumers? If so, please provide a description.

  **A:** We welcome interested third parties to contact us directly to discuss extending the dataset. Generally, adding new data involves using our app and following the recording guidelines to collect data. We also provide scripts on GitHub for data processing, ensuring that processed data conforms to the existing format and can be directly merged. We plan to implement a mechanism to comprehensively validate new technical contributions applied to the dataset and establish a leaderboard. Any extensions, enhancements, or contributions will be documented on GitHub.

- **Any other comments?**

  **A:** No.

# 8 Authorstatement

As the authors, we hereby affirm that we shoulder all accountability in any event of rights violations pertaining to the data collection or related undertakings, and will promptly enact necessary measures when required, such as purging data associated with such complications.

# 9 Consent Form of Participants

Within the succeeding sections, we delineate the Participant Information Sheet and Consent Agreement. Every participant accorded their signatures on the Consent Agreement prior to their involvement in this study.