# OpenReview forum: "Language Without Borders: A Dataset and Benchmark for Code-Switching Lip Reading"
_NeurIPS.cc/2024/Datasets_and_Benchmarks_Track — NeurIPS 2024 Track Datasets and Benchmarks Poster_

### Official Review · Reviewer_Rsda · 2024-07-19

**Rating:** 6
**Confidence:** 4

**Review:**

Overall, this paper introduces an interesting and particularly relevant dataset. The dataset is the first dataset to contain both Chinese and English videos for lip reading, is among the largest existing CN lip reading datasets, and is the first dataset to explore code switching in the lip reading space. Indeed, this paper opens up the potential for research into multilingual lip reading, which previously could only be explored with semi-synthetic data created from multiple monolingual datasets.
In terms of licensing, the dataset will be made publicly available under a CC-BY-4.0 license, with both the raw videos and transcripts available under that license, making for strong potential for re-use.

**Strengths:**

The key strength of this paper lies in the novelty and design of the dataset. This is the first (and as far as I am aware, only) dataset for code switching in lip reading, which makes it a valuable contribution as a benchmark for lip reading systems, and for opening the door to future research in multilingual lip reading. This paper also introduces and evaluates several baseline methods, which, while relatively weak in overall performance, demonstrate how existing lip reading techniques can be combined with existing code-switching ASR techniques to build code-switching specific models. The task is interesting as well, as lip reading is of great interest in the audio-visual-language communities.

**Additional Feedback:**

Overall, while the paper is a strong theoretical contribution, the paper's clarity could be greatly improved, and the organization of the paper, and the analysis of the baselines chosen are holding the paper back. If the paper focused on a single baseline method, along with a detailed discussion of the dataset itself, and some of the limitations, I think the paper would be stronger, however as it stands, I think that the data itself may be interesting enough to warrant an acceptance despite the analytical flaws. If some of the citations and motivations could be corrected and expanded, I would be open to changing my scores.

**Clarity:**

A lot of the paper is spent focusing on the analysis of the benchmark methods, particularly (mostly outdated) methods such as RNN/CNN based approaches. It would, in my opinion, have been better to focus several of those pages on the data collection process, motivation, and methodologies that appear in Section 1, 3, and 7 of the supplementary materials.  The analysis/code-switiching models introduced in Section, are, as far as I am aware, not particularly noteworthy beyond their application to the novel dataset, so I would prefer seeing more analysis of dataset composition, or details on dataset-specific phenomena (such as those in section 5.3) compared to detailed baseline methods discussions (which could be moved to the supplementary material).

**Correctness:**

The dataset construction appears to be correct and complete. Benchmarks appear reproducible, however are not particularly clearly motivated. In general, however, the dataset was collected in a valid, and ethical way. The dataset itself appears of high quality, with a high likelihood of having good transcription accuracy, and the dataset is high-information for the lip reading/code-switching task.

**Documentation:**

The dataset card presented in the supplementary material is complete, and contains sufficient information on the organization, availability and maintenance, and ethical and responsible use for the dataset.

**Ethics:**

I have no ethical concerns with this paper.

**Limitations:**

The limitations are discussed in the paper, with societal impact discussed in the appendix. While the paper does suggest some potential limitations such as anonymization, and variation in lip movements (which are inherent to lip reading with code switching), the paper would benefit from additional discussion on societal impacts such as:
- Lip reading privacy concerns in surveillance applications and issues with consent in lip reading models available in public spaces.
- The limitations of inaccurate interpretation and ethical concerns thereof
- Language biases (CN/English)

and potential dataset limitations such as:
- Demographic diversity (only 62 speakers collected in CN)
- Language proficiency (dataset speakers are proficient, but may not be the case for most applications)
- Sentence complexity and context variety
- Lack of video variation (controlled environment vs. in the wild)

**Opportunities For Improvement:**

While the paper is relatively strong, there are several opportunities for improvement in terms of novelty/contribution:
- **Comparison to monolingual dataset blending**: On L73, the paper claims that it is inaccessible to directly blend datasets from different languages to form a multilingual lip reading dataset, but this is not supported by either citations, or further motivation in the text. It would be nice to see a detailed comparison between these approaches in the paper, as it seems relatively sensible to try to combine datasets in these ways.
- **Baseline motivations/method architectures**: While the baselines are sufficient, the architectural choices are poorly motivated. For example, there is no indication as to why ShuffleNet is chosen as the feature front-end, or why the particular set of methods in section 5.2 were chosen. Further, section 5.2 would benefit from additional citations into the code switching ASR literature, since none of these methods are backed (currently) by any kind of citation or evidence that they would perform well.

**Relation To Prior Work:**

The relation to the prior work is fairly weak in this paper. While Table 1 does summarize existing dataset for lip reading, the related work section itself does not do an excellent job at placing this paper within the larger lip reading literature, or within the ASR code switching literature. I think in this case, it's a bit challenging, since this paper is the first to combine both, but it would be nice, for example, to have citations for the code-switching methods used in section 5.3, and a discussion of how code-switched ASR is different from standard ASR, and some of the techniques applied there. It would also be nice if there was a discussion on multilingual lip reading models, for example, models like [1], can apply to code-switched language.


[1] Ma, Pingchuan, Stavros Petridis, and Maja Pantic. "Visual speech recognition for multiple languages in the wild." Nature Machine Intelligence 4.11 (2022): 930-939.

**Summary And Contributions:**

This paper introduces CSLR (Cross-Linguistic Code-Switching Lip Reading), a bilingual (english-chinese) dataset for lip reading in code-switched scenarios. The dataset contains ~85K facial videos, and their associated transcriptions. The paper further provides a baseline pipeline for training which achieves EN WERs on the dataset as low as 49.05% on code-switched data, and CER as low as 20.25% on monolingual CN and WER as low as 22.66% on monolingual EN data from CSLR.

---

> ### Author Rebuttal · Authors · 2024-08-17
>
> **Q1: Blend datasets from different languages.**
>
> **Response:** Thanks for your insightful comments. In our opinions, directly blendding datasets from different languages is not suitable to build code-switching ASR datasets，which can be explained as follows:
>
> Code-switching typically occurs when words or phrases from other embedded languages appear at different positions within sentences of the primary language. Such datasets are extremely scarce, even in automatic speech recognition (ASR) tasks. The ASRU dataset is created using a mobile recording strategy, with samples like “我今天要去买一个 iPhone” and “Jeff 是一个很 sensitive 的学生.”
>
> Using synthetic methods to concatenate lip reading videos from different datasets, speakers, and languages presents significant challenges and does not conform to natural language patterns, resulting in low data quality.
>
> First, differences in lip morphology, skin color, head posture, and other environmental conditions between speakers can create substantial discrepancies at the seams of the concatenated clips, reducing feature learning efficiency.
>
> Second, it is difficult to find samples that adhere to the natural sentence order for concatenation. For example, synthesizing “Jeff 是一个很 sensitive 的学生” requires locating the segments “jeff,” “是一个很,” “sensitive,” and “的学生,” and precisely timing the splits and joins.
>
> Our goal is to provide high-quality sentences that reflect more natural code-switching scenarios, which is why we chose to create the dataset through recorded data.
>
> **Q2: Baseline motivations/method architectures.**
>
> **Response:** ResNet and ShuffleNet v2 are both widely used front-end feature extraction networks in the field of lip reading, applicable to both word-level and sentence-level lip reading [23], as well as various downstream tasks.
>
> ResNet offers enhanced representation discrimination at the cost of increased parameter count and computational load. **In contrast, ShuffleNet, as described in [23], has 5× fewer parameters and 12× fewer FLOPS than ResNet**, while only incurring a minimal performance loss, resulting in a significant boost in inference speed. This makes ShuffleNet particularly suitable for lip reading models designed for mobile devices where inference speed is critical [24][25].
>
> The pre-trained ShuffleNet on lip reading datasets also aids other downstream tasks, such as real-time audio-visual speech enhancement. In these tasks, ShuffleNet is used as a visual front-end to efficiently extract features and fuse them with audio signals, enabling real-time inference [27]. The trend towards lightweight network designs is increasingly prominent in current research [28], with advancements allowing these networks to achieve recognition accuracy comparable to or even better than ResNet.
>
> Based on the above considerations, we opted to use ShuffleNet as the front-end network and compared the recognition accuracy of using **ShuffleNet versus ResNet** in our dataset, **as shown in Table 2 of the PDF.**
>
> We compared the accuracy of ShuffleNet and ResNet under the Conformer and Vanilla CTC settings. Although the results indicate that ResNet achieves higher accuracy, it also requires more training resources.
>
> **Q3: Why the particular set of methods in section 5.2 were chosen.**
>
> **Response:** We will add a detailed explanation in the final version regarding the motivation behind the architecture choices and the relationships between different architectures.
>
> Since there has been no prior research on code-switching lip reading, we need to adapt methods from code-switching speech recognition to lip reading. Initially, the basic Vallina CTC framework [10][29] resembles the training framework for monolingual recognition and consists of a single-branch recognition network.
>
> This framework directly applies a consistent CTC calculation method without distinguishing between monolingual and code-switching (CS) samples when computing loss based on predictions and labels. The one-hot representation of labels includes both BPE segmentation for English and character IDs for Chinese. This framework overlooks the unique characteristics of different languages and is not effective for code-switching tasks.
>
> Building on the Vallina CTC model, some researchers have proposed the Bi-Encoder CTC framework [30], which expands the single-branch recognition network into a dual-branch system. Each branch initializes parameters using monolingual recognition networks to extract features with different language preferences. Features with monolingual language bias are concatenated and fed into a classification head for category prediction, supervised using the same loss calculation strategy as Vallina CTC.
>
> However, the supervision strategies of both the Vallina CTC and Bi-Encoder CTC frameworks fail to adequately distinguish the characteristics of monolingual data. The Language Aware Training framework [10][11][31][32][33] addresses this by adding monolingual supervision to the mixed loss calculation, enhancing language awareness. Unlike previous methods, this framework incorporates an additional monolingual loss term that masks the one-hot representations of other languages in the predictions and labels, only computing the differences for the specific language, thus providing independent supervision for each language.
>
> The Language Adaptive Mixture-of-Experts (MoE) framework [34] aims for a better integration and improvement of structure and supervision strategies. Unlike the previous dual-branch and single-branch methods that separately learn features, MoE employs dynamic gating networks for more effective feature integration [35], adding language-adaptive supervision to the gating layers [36]. This approach offers improved feature adaptability and better incorporation of monolingual information compared to the dual-branch architecture of Bi-Encoder CTC and the additional monolingual supervision strategy of Language Aware Training.

---

> > ### Author Rebuttal · Authors · 2024-08-17
> >
> > **Q4: Benchmarks are not particularly clearly motivated.**
> >
> > **Response:** Thanks for your comment. The motivation behind our proposed dataset is threefold:
> >
> > **1**. With the advancement of globalization, there is an increasing number of multilingual individuals who often switch between different languages while speaking.
> >
> > **2**. Certain concepts or terms can only be accurately expressed in specific languages, necessitating their inclusion within one's primary language.
> >
> > **3**. Additionally, there are specific groups of professionals, such as researchers whose native language is not English, for whom such language switching is unavoidable in communication.
> >
> > We will emphasize this content in the next version of the paper.
> >
> > **Q5: A lot of the paper is spent focusing on the analysis of the benchmark methods.**
> >
> > **Response:** We will take your suggestion and include this section in the supplementary materials. Thank you for your feedback!
> >
> > **Q6: More analysis of dataset composition, or details on dataset-specific phenomena.**
> >
> > **Response:** Thank you for your suggestions. We will move the model section to the supplementary materials and include an analysis of the dataset composition in the main text.
> >
> > Our dataset consists of 300 sentences as ground truth labels, including 100 sentences in Chinese, 100 sentences in English, and 100 code-switching sentences. These sentences are widely represented across various domains, as shown in Figure 2 of the main paper. Together, they depict 100 code-switching scenarios, with each scenario described using pure Chinese, pure English, and code-switching. We replaced words and phrases that are prone to code-switching in the primary language with embedded language equivalents and adjusted their order to better conform to natural language norms.
> >
> > This not only makes the dataset more reflective of actual code-switching scenarios but also enhances the recording quality of the speakers.
> >
> > To facilitate the calculation of CTC loss, we applied BPE (Byte Pair Encoding) segmentation to the vocabulary of all sentences.
> >
> > For the English portion, we directly used BPE to generate subwords, while for the Chinese portion, we adopted a strategy where each Chinese character represents one subword.
> >
> > Then we merged the subwords from both the Chinese and English portions, adding start symbol <s>, end symbol <e>, and unknown character <unk> to form the final subword set, consistent with previous work.
> >
> > This segmentation approach not only facilitates the expansion of the vocabulary without altering the existing one but also ensures that the subwords from the Chinese and English portions do not overlap. This makes it easier to implement Language Adaptive Training that retains only single-language subwords in the future.
> >
> > During the decoding process, our BPE segmentation strategy splits the English words while leaving the Chinese words intact, resulting in a higher proportion of English subwords. Consequently, the network needs to correctly predict all subwords of a word during the decoding phase to achieve accurate decoding.
> >
> > **Q7: Compared to detailed baseline methods discussions.**
> >
> > **Response:** Many thanks for your concern. We have provided a detailed response to this issue in our reply to Reviewer Rsda Q2. We have included this section in the supplementary materials based on your suggestion.
> >
> > **Q8: The relation to the prior work is weak in this paper.**
> >
> > **Response:** Thank you for your suggestions. We have added citations to Section 5.3 and provided additional explanations regarding the relationships between the methods; please refer to rsda Q3 for details.
> >
> > Single-language ASR [38] and VSR [39] have a long research history, where individual datasets are used to train single networks that can only recognize one language. Additionally, training multilingual ASR generally involves combining multiple monolingual datasets to create a multilingual dataset [40].
> >
> > Lip reading follows a similar paradigm to ASR, involving either the merging of datasets from different languages [40] or fine-tuning on multiple datasets after pre-training on large-scale datasets [6]. However, in both single-language and multilingual ASR, individual samples contain only one language and cannot handle code-switching samples, where different languages are used interchangeably. The goal of code-switching ASR is to extend ASR models to process different languages within a single segment of speech [5][10][29][30].
> >
> > Methods for lip reading code-switching have not been explored. This area presents greater challenges than code-switching ASR because the ambiguity of lip patterns is more severe than that of speech patterns, making it more difficult to distinguish between Chinese and English patterns and accurately decode code-switching sentences.
> >
> > Furthermore, there is currently no dataset available for training code-switching lip reading recognition networks.To ensure the production of a high-quality dataset, we have opted to hire volunteers and use high-definition cameras for recording.
> >
> > The limitations of our dataset may include: 1) the need to find more volunteers to record code-switching samples in other languages (e.g., French, Japanese), and 2) further expanding the diversity of sentences and adding longer recording samples (such as reading a segment of a code-switching text). We will focus on addressing these issues in our future work.

---

> > > ### Author Rebuttal · Authors · 2024-08-17
> > >
> > > **Q9: Additional discussion on societal impacts.**
> > >
> > > **Response:** We appreciate the reviewer's detailed feedback and valuable suggestions. We acknowledge the importance of expanding our discussion on societal impacts and potential limitations. To address this, we will make the following revisions to our manuscript:
> > >
> > > **1.Privacy Concerns and Consent:**
> > >
> > > We are deeply concerned about the privacy issues related to lip reading technology in surveillance applications and the consent required for using models in public spaces. We advocate for the introduction of stringent privacy protection measures, such as data encryption and anonymization. Furthermore, it is essential to obtain explicit informed consent when collecting data in public environments to adhere to ethical standards and safeguard individual privacy.
> > >
> > > **2.Inaccurate Interpretation:**
> > >
> > > Inaccurate interpretations of lip reading technology can lead to misunderstandings and the dissemination of misinformation, resulting in significant ethical issues. We advocate for the integration of contextual information and other auxiliary technologies in practical applications, and, where possible, the incorporation of audio information.
> > >
> > > Additionally, considering the speaker's tone and facial expressions is essential to enhance the accuracy of interpretations. We also call for the establishment of clearer usage guidelines and accountability mechanisms to ensure that the consequences of technology misuse can be effectively managed.
> > >
> > > **3.Language Biases:**
> > >
> > > Our dataset primarily encompasses Chinese and English, which may lead to a lack of generalizability in applications involving other languages and multilingual environments. In the future, we plan to expand the linguistic range of our dataset to explore inherent biases within the model and consider how these biases affect the generality and fairness of our system.
> > >
> > > For example, there is a language bias between Chinese and English: under the same amount of data, the same model, and the same hyperparameters, the word error rate (WER) for English is higher than the character error rate (CER) for Chinese. For further explanation and details on this issue, please refer to our response to Reviewer o2C4 Q2.
> > >
> > > **4.Demographic Diversity:**
> > >
> > > The current dataset only includes 62 speakers, all of whom are native Chinese speakers with English as their second language, which may not adequately represent the diverse characteristics of the global population. We recognize that increasing the number and diversity of speakers in the dataset is crucial for enhancing the model's applicability. Future data collection efforts will focus on expanding the variety of languages, ethnicities, and other demographic factors among the speakers.
> > >
> > > **5.Language Proficiency:**
> > >
> > > We will highlight that our dataset speakers are proficient, but this may not reflect the proficiency levels of users in real-world applications, potentially affecting the model's performance.
> > >
> > > **6.Sentence Complexity and Context Variety:**
> > >
> > > We recognize that the complexity of sentences and the limited variety of contexts in the dataset may impact the model's generalization ability. The current dataset includes a vocabulary that covers various aspects such as food, learning, work, hobbies, activities, daily life, and internationalization. The training and testing samples consist of both monolingual and bilingual alternations, with the sample types being unknown during testing. This design aims to increase the diversity of sentence structures and the complexity of contexts, thereby better simulating real-world communication situations.
> > >
> > > **7.Video Variation:**
> > >
> > > The current dataset is primarily collected using smartphone front-facing cameras, which are among the most widely used smart devices today. This makes data collection via smartphones extremely convenient and universal. Users can record videos in natural environments, making the data collection process more flexible and cost-effective.
> > >
> > > The smartphone perspective closely resembles how people communicate with others in their daily lives, so lip reading data collected through smartphones better reflects real-world application scenarios. This can enhance the model's applicability and performance in everyday situations, such as video calls with family and friends in noisy environments, as well as video content on social media.
> > >
> > > In the future, we plan to collect more data under various conditions, including different lighting backgrounds and dynamic environments, to improve the model's performance in complex and changing settings.

---

> > ### Author Rebuttal · Authors · 2024-08-23
> >
> > **Q10: Training Speed Comparison Between ResNet18 and ShuffleNet Front-end.**
> >
> > **Response:** With the back-end model configured as a Conformer, the model with a ResNet18 front-end requires 20 minutes and 2 seconds to run one epoch, while the model with a ShuffleNet front-end takes only 7 minutes and 33 seconds, making the training speed of the latter approximately 2.65 times faster than the former.
> >
> > **Q11: Discussion on Multilingual Lip Reading Models.**
> >
> > **Response:** We would like to clarify that the paper titled *"Visual Speech Recognition for Multiple Languages in the Wild"* published in *Nature Machine Intelligence* in 2022, essentially employed the same lip reading backbone, but trained and tested it separately on different monolingual lip reading datasets. This work did not achieve a model checkpoint capable of recognizing both multiple monolingual languages and code-switching lip reading scenarios simultaneously.

---

> ### Author Response · Authors · 2024-08-25
> **Detailed Response to Reviewer Rsda's Comments and Request for Further Feedback**
>
> **Dear Reviewer Rsda,**
>
> &emsp;Thank you again for your great efforts and valuable comments. We have carefully addressed all the concerns in detail. We hope you find the response satisfactory, similar to the other reviewers. **As the discussion phase is about to close, we are very much looking forward to hearing from you about any further feedback.** We will be very happy to clarify any additional concerns, should they arise.
>
> **Best regards,**
>
> **The Authors**

---

> > ### Comment · Reviewer_Rsda · 2024-08-25
> > **Response**
> >
> > I have read the review and adjusted my score accordingly.

---

> > > ### Author Response · Authors · 2024-08-26
> > > **Sincere Thanks for Reviewer Rsda's Feedback**
> > >
> > > We appreciate your positive feedback and for informing us about the score. Your insightful and constructive comments have been invaluable. Your involvement has significantly improved the quality of our work.

---

### Official Review · Reviewer_A5e4 · 2024-07-24
**Lip reading has advanced greatly, helping those with speech impairments and in noisy settings. However, current datasets do not cater to bilingual users, and research on code-switching in lip reading is minimal. To fill this gap, the CSLR dataset was developed, featuring Chinese and English video samples. This dataset includes 85,000 videos from 62 bilingual speakers fluent in both languages. SPIGA is utilized to detect the lip region of interest (ROI), crucial for subsequent model development.**

**Rating:** 7
**Confidence:** 4

**Review:**

Quality:
The data collection process addresses a critical gap between multilingual and code-switching utterances. The collected data is also carefully analyzed for quality check by the same 62 annotators who are proficient in both the languages.

Originality:
1. This work is the first work on lip reading dataset in code-switching, especially in Chinese-English, and also conducts thorough experimentation on this use case.

Significance:
1. With the widespread usage of social media, more studies on code-switching especially in multimedia are required. This work puts forth a clean dataset for code-switching lip reading while not compromising on the monolingual counterparts.
2. Language Routing MoE performed the best and adapting the language labels to define cross-entropy loss for MoE is interesting. This formulation can also be used for other code-switching tasks such as translation and entity detection.

Pros:
1. The dataset is carefully designed such that while performing well on the CS parts, the models can still be evaluated on each individual language counterpart as well.
2. Careful curation of high-quality samples in the dataset from manual examination.
3. Developed LipGather App to record and store data on cloud. Since the setup is such that it can be remote, future work can potentially collect data from more languages using this app.

Cons:
1. This is a major concern: The dataset is divided into 3-second frames, limiting the use of contextual information for word prediction in lip reading. This deliberate fragmentation complicates the task, and the reasoning behind this design choice is unclear and counterintuitive.
2. The diversity of the dataset considering that it is only 300 sentences is unclear.
3. The instructions for rigorous screening performed for data quality filtering are not clearly mentioned. This makes it hard to understand which kinds of erroneous elements existed in the original dataset that needed to be filtered.
4. In the language aware training, it is unclear how we are getting the language label for masking.

Missing citations:
For the claim of multilingual lip reading datasets would not be enough for code-switching in Section 2.1:
[1] Multilingual Large Language Models Are Not (Yet) Code-Switchers
[2] Are Multilingual Models Effective in Code-Switching?

Missing reference for IFQA.

**Strengths:**

1. Lip reading tools are immensely helpful for speech impaired individuals and is societally very beneficial. This tool can significantly improve their quality of life by providing a vital means of understanding others, especially in environments where auditory cues are limited or unavailable.
2. Thorough examination among different architecture choices is conducted that can help with  model selection for subsequent works.
3. Not just with prior SOTA models as is but these architectures are also adapted well and compared with code-switching based methods includes vallina CTC, bi-encoder CTC, language id training, language routing with MoE. This gives valuable insights to the communities on adopting existing models to this switching setup.

**Additional Feedback:**

Minor corrections:
Line 52: “valid yet high-quality” → “valid and high-quality”
Line 58/59: “the challenging of CSLR with” – something is missing here, perhaps, the challenging nature of
Line 105: “may not only derive from” → “may not only be derived from”
Line 138: “that takes account” → “that takes into account”
Line 145: “lip reading, We” → “lip reading, we”
IFQA score in Fig 2 needs to be directed to appropriate section 4.2

**Clarity:**

The paper is well written and easy to follow, although the presentation could benefit from minor improvements.
Firstly, the experimental setup is clearly explained in two segments for the monolingual and code-switching setups. Some concepts are introduced before explaining what they are and not making appropriate citations. They can be fixed, please see additional feedback.

**Correctness:**

The authors mention in Introduction and section 3.3 that they introduce / customize new metrics. However, code-switching has been evaluated on word error rate and character error rates in the prior work as well [1,2] and many more. Its better to tone down this claim.

It is unclear if the LipGather App that is developed as a part of this work is going to be released. Since this is not mentioned as one of the contributions, its alright either ways but making this clear would be helpful.

Refs:
[1] Transliteration Based Approaches to Improve Code-Switched Speech Recognition Performance
[2] Code-Switching Automatic Speech Recognition for Nursing Record Documentation: System Development and Evaluation

**Documentation:**

The checklist mentions that the URL to code and dataset is provided but the git link does not seem to contain the data yet.

**Limitations:**

The inadequacy of anonymization is well written and explained. Visual similarity of auditorily similar sounds is challenging even in monolingual cases. So unlike the authors mentioned that it makes code-switching challenging, I think this particular limitation, such as the visual signals of sounds like ‘p’ and ‘b’ being similar would be generally challenging for lip reading.

**Opportunities For Improvement:**

The dataset is split into smaller 3 second frames. This hinders utilizing the context in predicting the next word which is a natural part of lip reading. But breaking this context deliberately makes this hard and this design choice in dataset creation is not clearly explained with reasons as it seems counterintuitive.

Explain clearly about the annotation instructions for quality filtering and whether the annotations also include a language id.

**Relation To Prior Work:**

Comparison and contrasts to prior work is well presented on the datasets side. Since the benchmark is the main contribution, this comparison is important for the paper. However, as the model architectures have been used in prior work, like Conformer or MSTCN, referencing them along with mentioning their relative performances on prior tasks would give a holistic understanding of where these components have been used.

**Summary And Contributions:**

Lip reading has progressed significantly, benefiting speech-impaired individuals and noisy environments. Existing datasets lack support for bilingual users, and code-switching in lip reading is under-researched. To address this, the CSLR dataset, containing Chinese and English video samples, was created. It includes 85k videos from 62 bilingual speakers proficient in both languages. They use SPIGA for detecting lip-ROI that is used for further models.
Notable contributions:
1. The benchmark totals to 71 hours of high quality code-switching lip-movement data. The dataset is collected from 300 high-quality Chinese-English sentences from questionnaires.
2. The authors experimented with SOTA lip reading models from monolingual settings to benchmark their performances on CSLR with different front and backbone architectures.
3. Not only this, they also adapted the typical code-switching architectures such as Vallina CTC, Bi-Encoder CTC, Language Aware Training, and Language-Routing Mixture of Experts to this task and observed that MoE performs the best among them.

---

> ### Author Rebuttal · Authors · 2024-08-17
>
> **Q1: The dataset is divided into 3-second frames, limiting the use of contextual information for word prediction in lip reading.**
>
> **Response:** We admit that unifying the dataset to a duration of 3 seconds may indeed affect the utilization of long-context information. However, during the actual training process of the lip reading network, varying the lengths of video samples can lead to instability in network training. For this reason, we have aligned all samples to a uniform length of 3 seconds, which is similar to the processing methods used in other datasets, such as GRID and OuluVS2, where samples are standardized to 3 seconds and a fixed number of frames, such as 50 frames, for training and testing the network.
>
> In fact, using a 3-second video setting can accommodate over 30 one-hot encodings, thus allowing for some degree of contextual information to be leveraged for decoding the spoken content. In our future research, we plan to attempt collecting samples where speakers read a code-switching paragraph to evaluate the impact of long-context information on code-switching lip reading.
>
> **Q2: The diversity of the dataset considering that it is only 300 sentences is unclear.**
>
> **Response:** Our dataset consists of carefully designed sentences, as shown in Figure 2, which encompass seven common scenarios in daily life where code-switching frequently occurs. These sentences are crafted to include a variety of challenging Chinese and English words and characters that are prone to confusion, allowing us to evaluate the model's performance on code-switching.
>
> Additionally, the sentences feature multiple instances of code-switching along with a diverse ratio of Chinese and English content.
>
> In our future work, we plan to create a dataset that includes a broader range of sentences to further enhance the benchmarking capabilities for code-switching lip reading.
>
> **Q3: The instructions for rigorous screening performed for data quality filtering are not clearly mentioned.**
>
> **Response:** Thanks for your constructive comments. The relevant responses can be found in Reviewer A5e4 Q7.
>
> **Q4: In the language aware training, it is unclear how we are getting the language label for masking.**
>
> **Response:** The purpose of language-aware training is to provide monolingual supervision signals. During the training process for code-switching, it is possible to encounter situations where a label contains both Chinese one-hot encodings and English one-hot encodings. The method for creating monolingual supervision labels involves marking all other languages as except for the target language, while retaining only the one-hot encoding of the target language to compute the monolingual loss. For example, the Chinese mask label for the phrase “我喜欢ham bur ger” would be “我喜欢<unk><unk><unk>”, while the English mask label would be “<unk><unk><unk>ham bur ger”.
>
> **Q5: Missing citations: For the claim of multilingual lip reading datasets would not be enough for code-switching in Section 2.1.**
>
> **Response:** Thank you for pointing out the missing citations. These references are crucial to support our claim regarding the inadequacy of multilingual lip reading datasets for code-switching. We have incorporated these references into Section 2.1 to substantiate our argument.

---

> ### Author Rebuttal · Authors · 2024-08-17
>
> **Q6: Missing reference for IFQA.**
>
> **Response:** Sorry for the oversight regarding the missing reference for Interpretable Face Quality Assessment (IFQA)[46]. We have ensured that the reference for IFQA is included in the manuscript. Thank you for bringing this to our attention.
>
> **Q7: Explain clearly about the annotation instructions for quality filtering and whether the annotations also include a language id.**
>
> **Response:** Our expert filtering strategy primarily aims to remove low-quality and erroneous samples. A sample is marked as low quality if the speaker exhibits significant head movement, moves out of the camera frame, or if there are objects obstructing the lip area during speech.
>
> We employ a three-class language identification labeling system that distinguishes between monolingual Chinese, monolingual English, and code-switching samples. This labeling approach has been widely validated in automatic speech recognition to enhance the model's performance in learning code-switching.
>
> **Q8: The authors mention in Introduction and section 3.3 that they introduce / customize new metrics. It’s better to tone down this claim.**
>
> **Response:** Thanks for your valuable feedback. We have toned down our claim to reflect the contributions of prior research more accurately, ensuring that our manuscript is more rigorous.
>
> **Q9: Some concepts are introduced before explaining what they are and not making appropriate citations.**
>
> **Response:** Thanks for your constructive feedback. We have reviewed the manuscript and made the necessary corrections to ensure that concepts are introduced with proper explanations and appropriate citations.
>
> **Q10: The checklist mentions that the URL to code and dataset is provided but the git link does not seem to contain the data yet.**
>
> **Response:** Thanks for bringing this to our attention. Due to the limited space in the main text, we have provided the download method for the dataset in Section 4 of the supplementary material titled "Access to Dataset and Benchmark."
>
> In addition, the supplementary material includes detailed information on the recording pipeline, dataset structure, training and model hyperparameters, societal impact, license, and dataset sheet, among other relevant details. We hope this addresses your concern and provides the necessary information.
>
> **Q11: Minor corrections.**
>
> **Response:** Thank you for your detailed feedback and suggestions for minor corrections. We have carefully reviewed your comments and have made the necessary corrections in the manuscript as follows:
>
> ① Line 52: Changed “valid yet high-quality” to “valid and high-quality.”
>
> ② Line 58/59: Corrected "the challenging of CSLR with" to "the recognition difficulty of CSLR with".
>
> ③ Line 105: Corrected “may not only derive from” to “may not only be derived from.”
>
> ④ Line 138: Adjusted “that takes account” to “that takes into account.”
>
> ⑤ Line 145: Corrected “lip reading, We” to “lip reading, we.”
>
> ⑥ Directed the IFQA score in Fig 2 to the appropriate section 4.2.
>
> We appreciate your insights and believe that these changes have improved the clarity and accuracy of our manuscript.

---

### Official Review · Reviewer_8wJh · 2024-07-25
**Pioneering code-switching lip reading dataset**

**Rating:** 9
**Confidence:** 5
**Correctness:** The results seem correct.
**Clarity:** Yes, the paper is well written.

**Review:**

In the paper, a benchmark in the form of a novel dataset for visual speech recognition that contain bilingual phrases is introduced. Also, the performance of several representative methods is examined.  Pros: *Pioneering code-switching lip reading dataset, * Comparative evaluation of approaches (monolingual and code-switching); Cons:  *Unclear division of data samples into training/testing groups, *Limited benchmarking.

**Strengths:**

*Novel dataset
*Experiments with recent approaches.

**Additional Feedback:**

*ASRU2019 [37] is not included in the Table 1 but mentioned in Section 2.2.

**Documentation:**

The details on data collection and organization are sufficient.

**Ethics:**

There are no ethical concerns related to this study.

**Limitations:**

* Such a large dataset can be used to thoroughly examine representative methods.

**Opportunities For Improvement:**

1.	The main text should contain the information of data division into testing and training examples. Why was 10-fold cross validation not used? Also, the experiments varying the number of training examples should be performed and the results of different methods reported to show their robustness
2.	Can the introduced dataset be used to improve monolingual lip reading recognition? Please elaborate.

**Relation To Prior Work:**

The limitations of previous studies are discussed.

**Summary And Contributions:**

In the paper, a novel dataset for code-switching lip reading purposes is introduced along with evaluation of widely-used approaches.

---

> ### Author Rebuttal · Authors · 2024-08-17
>
> **Q1: The main text should contain the information of data division into testing and training examples.**
>
> **Response:** Thank you for your feedback. Similar to the partitioning rules used in other lip reading and speech recognition studies, we will include the dataset partitioning rules in the main text. We will divide the dataset based on the recorded sentences, ensuring that the subwords encountered during training include those found in the test set. This allows the test set to feature unseen combinations of subwords, enabling the decoding of words not previously encountered in the training set.
>
> This partitioning approach is random, meaning that there are multiple valid ways to create the splits. We randomly selected one set of test data from among several options that meet the criteria. This method of partitioning is effective for validating the model's decoding capability when faced with unknown code-switching combinations.
>
> **Q2: The 10-fold cross validation and varying the number of training examples.**
>
> **Response:** K-fold cross-validation is typically used to evaluate a model’s susceptibility to overfitting or selection bias, providing a more generalized assessment. In existing lip reading datasets [41][42][43], this strategy is rarely employed, as the datasets are generally large enough to provide a high-quality evaluation of model performance.
>
> To further enhance the evaluation capability of our dataset, we are experimenting with three random splits of training and testing data, ensuring that the subwords in the training set include those in the test set. This experiment is currently underway, and the results will be presented in the discussion section. Thanks for your patience.
>
> **Q3: Experiments should be conducted to change the number of training samples.**
>
> **Response:** Based on your suggestions, we conducted experiments by varying the number of training samples while keeping the test set constant, specifically by discarding 25% of the training samples. This experiment is currently in progress, and the results will be presented in the discussion section. Thanks for your patience.
>
> **Q4: Introduced dataset be used to improve monolingual lip reading?**
>
> **Response:** Our dataset can help improve the accuracy of monolingual lip reading to a certain extent, and there are several potential technical approaches to consider:
>
> **1.Unsupervised Pre-training:** Due to the high quality and large scale of our dataset, which includes both Chinese and English lip movement patterns, it can provide better mode variation for unsupervised pre-training. This can mitigate model overfitting and enhance the pre-training of monolingual lip reading networks.
>
> **2.Blending Datasets:** Our dataset offers high-quality monolingual samples. Furthermore, we can utilize the LAT[10] method to convert code-switching bilingual samples into monolingual samples for monolingual lip reading training. The masked segments of the other language can act as a form of data augmentation, thereby enhancing robustness. This approach can be integrated with other monolingual datasets to increase the data volume and improve the accuracy of monolingual lip reading.

---

> > ### Author Rebuttal · Authors · 2024-08-23
> >
> > **Q5 Experiments should be Conducted to Change the Number of Training Samples.**
> >
> > **Response:** We conducted an experiment using the ShuffleNet+Conformer+Vanilla CTC settings with 75% of the original training data, based on the train-test split described in the main paper, while keeping the test set unchanged. We also keep the consistent learning rate, number of iterations, and other hyperparameters. The results, as shown in Table 3 of the PDF, indicate that using 75% of the training data led to a 3.69% increase in CER, a 6.17% increase in WER, and a 4.60% increase in MER compared to using the full dataset.
> >
> > These results suggest that reducing the training data slightly impacts the model's performance, highlighting the challenges of achieving optimal fit in the code-switching lip reading task. However, it is noteworthy that the model retained a significant level of accuracy, demonstrating its robustness even with a smaller training set. This experiments also provided insights into potential overfitting issues, which can be more easily identified when less data is used.
> >
> > In response to the reviewer's suggestion, we recognize the importance of evaluating the model's generalization capabilities under data constraints, which is reflective of real-world applications. Moving forward, we plan to continue expanding the dataset and exploring advanced data augmentation and regularization techniques to further enhance the model's performance and generalization, even when data is limited.
> >
> > **Q6: Perform Cross Validation on the Dataset Proposed in This Paper.**
> >
> > **Response:** To further enhance the evaluation of our proposed benchmark, we conducted three-fold cross-validation experiments on the dataset introduced in this paper, using ShuffleNet+Conformer+Vanilla CTC settings. During the dataset split process, we ensured that the subwords in the test set were also present in the training set, and the test datasets in each split were entirely non-overlapping.
> >
> > The results, as shown in Table 4 of the PDF, indicate that the model's performance remained relatively consistent across the folds, with slight variations in error rates. These variations likely stem from differences in data distribution among the splits, particularly the presence of more challenging subwords or sentences in certain test sets. Overall, the results demonstrate the model's robustness, while also highlighting the importance of a well-distributed dataset for code-switching lip-reading tasks.
> >
> > We have incorporated these cross-validation results into our manuscript to address the reviewer's request for a k-fold dataset split and to provide a more comprehensive evaluation of our model.
> >
> > **Q7: ASRU 2019 Mandarin-English Code-Switching Speech Recognition Challenge.**
> >
> > **Response:** We sincerely appreciate your valuable comments. Since the code-switching speech recognition dataset was collected in a quiet room using a smartphone and does not include lip reading information, we did not discuss it in Table 1 of our manuscript. However, we will address this issue in the final submission and provide detailed information about the ASRU 2019 dataset [5].

---

### Official Review · Reviewer_o2C4 · 2024-07-25

**Rating:** 6
**Confidence:** 3
**Correctness:** Yes, largely so.
**Clarity:** Yes.

**Review:**

Pros:

1. Paper proposes a benchmark for a new task -- bilingual code switching lip-reading -- that they motivate to be a useful task as in bilingual conversational contexts, people generally employ and switch between languages to express thoughts.

2. The benchmark construction process is mostly adequately described and a mobile app is designed to facilitate collection. The benchmark is released through github as well.

3. The baselines considered comprise three prominent base architectural types -- CNNs, LSTMs and Transformers -- as well as code-switching models. Experiments and underlying methods are adequately described.

Cons/Opportunities for Improvement:

1. Currently, seeing the experimental results, I feel the current benchmark's complexity seems to arise more from the English word transcription rather than the code-switching aspect (in all tables the error rate of the English transcription, WER, is much higher than both CER and mixture of the two). Ideally one more language pair (e.g. French and English) would provide a more interesting and thorough account for bilingual code-switched lip reading. Currently, the dataset is limited to only Chinese and English languages which is one specific case of bilingual code-switching.

2. The human accuracy on this task is currently not provided. It will be useful to highlight the current performance gap of existing methods against humans. Further, a more detailed breakdown of the types of errors that occur or doing more detailed analysis of where current models struggle most can make the paper more complete (currently only a qualitative example is provided). E.g. are there certain phrases/words that have the most common errors, or are there certain chinese-english switch patterns/phrases where models struggle more?

3. Relatively minor: The zero/few-shot performances of current multimodal LLMs / Video-language models (e.g. GPT4, Gemini, VILA) may also be interesting to include to showcase challenge of the task and potentially increase impact of the paper.

**Strengths:**

Please see the pros section.

**Additional Feedback:**

NA

**Documentation:**

Yes, a github url is provided, and supplemental consists of more information.

**Ethics:**

For a research involving human subjects (especially face videos), the paper does not have (or perhaps I have missed) information regarding consent collection or institutional review board approval. Including this would be useful (a screenshot of the app consent page should be useful).

I am unsure if this is a significant ethics concern, and have currently put minor ethics concerns and seek opinion of other reviewers/AC.

**Limitations:**

Yes, there is a section on limitations in main paper and negative social impact in supplemental.

**Opportunities For Improvement:**

Please see Review section

**Relation To Prior Work:**

Related work section showcases differences from prior benchmarks and works.

**Summary And Contributions:**

The authors propose a bilingual lip reading benchmark termed CSLR which comprises face-view videos of participants talking in English and Chinese (interspersed) and requires models to annotate the spoken speech. The authors design a mobile application for data collection from 62 participants, and the dataset comprises 85,560 videos after filtering for erroneous cases. Authors finally analyze performances of existing lip-reading backbones and find performances are low across three considered evaluation metrics (Chinese character level, English word level and mixture of the two), besides doing qualitative error analysis.

---

> ### Author Rebuttal · Authors · 2024-08-17
>
> **Q1: The Word Error Rate (WER) for English transcription is higher than the Character Error Rate (CER) and Mixed Error Rate (MER).**
>
> **Response:** Thanks for your insightful question. From the experimental results, English word transcriptions have more complexity with high error rates, but such complexity is actually not  derived from our proposed code-switching benchmark. This phenomenon is reflected not only in code-switching lip reading but also in code-switching speech recognition [10][11]. This is because a Chinese character consists of a single glyph, while an English word can contain multiple characters.
>
> For example, in the code-switching sample "论文 COAUTHORS," the Chinese word "论文" consists of two tokens: "论,374" and "文,268." In contrast, the English word "COAUTHORS" is broken down into seven tokens: "▁C,25," "O,106," "A,104," "U,117," "TH,101," "OR,15," and "S,109."
>
> In English prediction, it is necessary to correctly predict all tokens of a word in order to output the correct word. In contrast, for Chinese characters, predicting just one token is sufficient to correctly identify the character. Therefore, due to the influence of linguistic characteristics, predicting a complete English word is generally more challenging than predicting a single Chinese character. This difficulty is evident not only in code-switching lip reading, but also in code-switching speech recognition [10][30].
>
> **Q2: Adding an additional language pair (e.g., French and English).**
>
> **Response:** Thank you for your valuable feedback. We fully agree on the importance of incorporating additional language pairs, such as French and English. Currently, our dataset focuses on code-switching between Chinese and English, which is a common scenario in real life[37]. We initially chose to work with Chinese and English in the early stages and aim to accumulate more experience and methodologies based on our preliminary results. In the future, we will collect lip reading data in various linguistic environments, including but not limited to combinations of French and English, to validate the universality and effectiveness of the proposed methods across different language contexts.
>
> **Q3: The human accuracy on this task.**
>
> **Response:** The Oxford University Artificial Intelligence Lab, the Google DeepMind team, and the Canadian Institute for Advanced Research (CIFAR) jointly released the lip reading program LipNet, which integrates deep learning technology, in 2016 [7,8]. On the GRID corpus, LipNet achieved an accuracy of 93.4%, surpassing both experienced human lip readers and the previous best accuracy of 79.6%. The researchers also compared LipNet's performance with that of hearing-impaired individuals who are proficient in lip reading. On average, these individuals achieved an accuracy of 52.3%, while LipNet's performance on the same sentences was 1.78 times this accuracy.
>
> We also invited three human lip readers to conduct experiments on our test set. The results will be supplemented during the discussion phase and will be included in the final manuscript. Thank you for your suggestions and patience. If you have any questions, we look forward to further communication during the discussion phase.
>
> **Q4: A more detailed breakdown of the types of errors.**
>
> **Response:** Thank you for your valuable feedback. We have conducted a statistical analysis of the erroneous samples and categorized the types of errors into three main groups.
>
> **The First Category** consists of **errors caused by the lip ROI visual dynamic pattern similarities** between Chinese and English, often referred to as difficult or confusing samples [45]. This type of error can be further divided into three subcategories:
>
> 1）Chinese misidentified as Chinese: Common examples include "开会" being misrecognized as "口水," "基线" being recognized as "想法," and "小组" being identified as "调周."
>
> 2）Chinese misidentified as English: Notable examples include "分析" misrecognized as "VISA," "基线" being identified as "TIMELINE," "GOOGLE" being recognized as "阅读," and "标志" being misidentified as "LUNCH."
>
> 3）English misidentified as English: Examples include "BACKUP" recognized as "BENCHMARK," "BIG" misidentified as "PIZZA," "WEEK" recognized as "WORK," and "NEXT" being misrecognized as "DOESN'T."
>
> **The Second Category** involves **spelling errors**, primarily concerning the incorrect spelling of English words or Chinese phrases. For instance, "BRAINSTORM" may be misrecognized as "BRAINSTARM," "CONFERENCE" as "CONFOREACE," and "最佳球员" as "最打球员."
>
> **The Third Category** refers to **code-switching errors**. We compiled a confusion matrix of predictions and labels **as shown in Figure1 of the PDF**. Since our task involves sentence-level recognition, we cannot achieve precise one-to-one mapping; thus, any prediction that includes the target word is considered correct, regardless of its position. We selected two pairs of easily confused code-switching words: COFFEE and 咖啡, and BUY and 买, for analysis.
>
> "COFFEE" had 95 samples correctly predicted, with 10 samples predicted as "咖啡". "咖啡" had 43 samples correctly predicted, with 0 samples predicted as "COFFEE". "BUY" had 55 samples correctly predicted, with 0 samples predicted as "买", while "买" had 54 samples correctly predicted, but 268 samples predicted as "BUY". We found that the prediction results for "COFFEE" and "咖啡" were better, possibly due to the significant differences in their lip movement patterns. In contrast, predicting BUY and 买 was much more challenging because their lip movement patterns are very similar and short, leading to a high likelihood of incorrect predictions.

---

> > ### Author Rebuttal · Authors · 2024-08-17
> >
> > **Q5: The zero/few-shot performances of current multimodal LLMs / Video-language models.**
> >
> > **Response:** Using large multimodal LLMs / Video-language models for code-switching research is a promising idea, but there are several technical issues that need to be addressed. Currently, video-language large models primarily utilize real-world videos along with corresponding textual descriptions [12][13][14] to learn the relationships between video and text. These videos typically include scenes from autonomous driving [15] or descriptions of movements between objects [16], as well as human actions [17][18].
> >
> > Current research indicates that video-language models can learn the relationships between scenes and actions in videos and their textual counterparts; however, this learning requirement differs significantly from the needs in lip reading, which involves the relationship between video and spoken content.
> >
> > The videos used in current video understanding tasks contain a wealth of semantic information embedded in the spatial structure of the frames (for example, recognizing a person playing beach volleyball at the seaside requires that almost all frames include the beach, the volleyball, and the person, along with the action of playing). This information can be effectively extracted without relying on context (adjacent frames) [19].
> >
> > This characteristic of the task is also reflected in the design of network structures, as contemporary action recognition models commonly use a single network responsible for spatial and temporal modeling (such as temporal shifts in spatial convolution [20] or a combination of spatial-temporal convolutions [21], or modeling the differences between frames [22]), with a relatively weak demand for temporal modeling.
> >
> > In contrast to previous video understanding tasks, the videos used in lip reading differ greatly from real-world scenes, as each frame focuses solely on the speaker's lips. It is challenging to infer spoken content based solely on a single frame of the lips; instead, the emphasis is on the changes in the lip area and the long-term context needed to infer complete sentences.
> >
> > This also necessitates that all existing word-level and sentence-level lip reading models incorporate a front-end model (such as 3D+2D ResNet or ShuffleNet) to extract short-term changes, while employing a back-end model (like GRU, Conformer, or MSTCN) for long-term context modeling.
> >
> > These task differences limit the application of advanced video-language models in lip reading tasks. However, we still have the opportunity to leverage large-scale pre-trained models to enhance the performance of lip reading.
> >
> > We use the GPT-4 model you mentioned to correct our prediction results, and the prompts we used are as follows:
> >
> > *We conducting research on code-switching. Please correct the spelling errors in the predicted text generated by my deep learning model, ensuring that the English words remain in English and the Chinese words remain in Chinese. Your input will be my predicted text. For each line of output, you should provide both the original text and the corrected text, separated by a semicolon. Each output should occupy one line. Here is an example: For the input '想 法 不 WARK', your output should be '想 法 不 WARK; 想 法 不 WORK'. Please strictly follow this format, without adding any extra content or numbering the outputs.*
> >
> > The prompts will be followed by our model's prediction results. We corrected the predictions using the **GPT-4o** model in batches of 50. The experiments were conducted with the predictions from a baseline setup using ShuffleNet + Conformer + Vanilla CTC. **The metrics before and after correction are shown in Table 1 of the PDF.**
> >
> > The corrections made by the GPT-4 model significantly improved the recognition accuracy of English words. Since words are composed of multiple subwords, any error can lead to incorrect decoding or omission of the word. The GPT-4 model effectively repairs these partial errors or omissions based on the already decoded parts of the words and contextual semantics, thereby enhancing the Word Error Rate (WER) accuracy.
> >
> > However, we also observed that GPT-4 performed poorly on the Chinese components in code-switching scenarios. This may be attributed to the independence of Chinese words; even if there were prediction errors in the characters, the model struggled to identify these mistakes. This difficulty in correction, or even the introduction of incorrect modifications, resulted in a decrease in the final Character Error Rate (CER).
> >
> > **Q6: The paper does not have information regarding consent collection or institutional review board approval.**
> >
> > **Response:** Before collecting facial videos from volunteers, we obtained their prior consent, and each volunteer signed an informed consent form, a template of which is attached. As a token of appreciation, we provided appropriate compensation to each volunteer.
> >
> > To protect the speakers' privacy, we only released the NPZ format data of the lip region of interest after landmark localization and cropping. We have provided only a limited number of full-face videos/images to give reviewers and other readers a visual understanding of our work.

---

> > ### Author Rebuttal · Authors · 2024-08-23
> >
> > **Q7: Human Accuracy On Our Dataset.**
> >
> > **Response:** Thanks for your patience. We recruited six students from our school who are proficient in both Mandarin and English, including three males and three females. These participants frequently use code-switching sentences in their daily communications. We selected 100 sentences from the sentence set of our dataset, dividing them into 70 for training and 30 for testing.
> >
> > During the training phase, the participants could only rely on the samples from these 70 sentences, along with their corresponding speech content labels, to undergo a certain level of lip reading training. Subsequently, they were tested by the aforementioned 30 test sentences.
> >
> > We compiled the prediction results of the six participants and conducted post-test interviews with them. All six participants reported that predicting speech content solely based on lip movements from the video was extremely challenging. Specifically, their average MER was 97.43%, the average WER was 110.48%, and the average CER was 90.00%. This may be due to the complexity of lip reading tasks for participants who have not undergone professional training.
> >
> > Building on this, we conducted an additional experiment where the participants were allowed to watch the test sentences along with their labels once before making predictions solely based on the video during the second attempt. We found that even under this simplified experimental setup, the accuracy of human lip readers remained significantly lower than that of the deep learning models we employed.
> >
> > These experimental results suggest that deploying code-switching lip reading methods in real-world environments holds significant promise.

---

> > > ### Author Rebuttal · Authors · 2024-08-24
> > >
> > > **Q8: Will LipGather-APP be made publicly available for researchers to use?**
> > >
> > > **Response:** The project of the data collection app has been made publicly available on GitHub at https://github.com/cslr-lipreading/CSLR/tree/main.

---

> ### Comment · Reviewer_o2C4 · 2024-08-24
>
> Thank you for the clarificafions and detailed rebuttal. My concerns have been largely addressed and authors should include additional results, human performances and limitations (that only a single language pair of chinese & english is considered) in the revised version. Authors should also limit conclusions/claims to this specific language pair rafher than general language pairs for code switching where appicable. I have raised my score.

---

> > ### Author Response · Authors · 2024-08-24
> > **Replying to Official Comment by Reviewer o2C4**
> >
> > We would like to express our sincere appreciation for your valuable feedback, which was instrumental in enhancing the quality of our manuscript. If you have any additional suggestions, comments, or requirements, please feel free to contact us.

---

### Author Rebuttal · Authors · 2024-08-17

**Many thanks for your detailed review and insightful feedback. Your suggestions have greatly enhanced the quality of our manuscript. We have thoroughly replied to all the issues mentioned in each review.**

**Q1: Will LipGather-APP be made publicly available for researchers to use?**

**Response:** The project of the data collection app has been made publicly available on GitHub at https://github.com/cslr-lipreading/CSLR/tree/main .


Below are the references in the response:

**[1]** Altieri, N A, Pisoni, D B, Townsend, J T. "Some normative data on lip-reading skills (L)." The Journal of the Acoustical Society of America, 2011, 130(1): 1-4.

**[2]** Le Scao, T, Fan, A, Akiki, C, Pavlick, E, Ilić, S, Hesslow, D, Castagné, R, et al. "Bloom: A 176b-parameter open-access multilingual language model." 2023.

**[3]** Emond, J, Ramabhadran, B, Roark, B, Moreno, P, Ma, M. "Transliteration based approaches to improve code-switched speech recognition performance." In 2018 IEEE Spoken Language Technology Workshop (SLT), pp. 448-455. IEEE, 2018.

**[4]** Hou, S-Y, Wu, Y-L, Chen, K-C, Chang, T-A, Hsu, Y-M, Chuang, S-J, Chang, Y, Hsu, K-C. "Code-switching automatic speech recognition for nursing record documentation: system development and evaluation." JMIR Nursing, 2022, 5(1): e37562.

**[5]** Shi, A, Feng, Q, Xie, L. "The ASRU 2019 Mandarin-English code-switching speech recognition challenge: Open datasets, tracks, methods and results." arXiv preprint arXiv:2007.05916, 2020.

**[6]** Ma, P, Petridis, S, Pantic, M. "Visual speech recognition for multiple languages in the wild." Nature Machine Intelligence, 2022, 4(11): 930-939.

**[7]** MIT Technology Review. "AI has beaten humans at lip reading." 2016. https://www.technologyreview.com/2016/11/21/69566/ai-has-beaten-humans-at-lip-reading/

**[8]** Assael, Y M, Shillingford, B, Whiteson, S, de Freitas, N. "LipNet: End-to-end sentence-level lipreading." arXiv preprint arXiv:1611.01599, 2016.

**[9]** Prajwal, K R, Afouras, T, Zisserman, A. "Sub-word level lip reading with visual attention." In Proceedings of the IEEE/CVF Conference on Computer Vision and Pattern Recognition, pp. 5162-5172. 2022.

**[10]** Tian, Y, Yu, J, Zhang, C, Weng, C, Zou, Y, Yu, D. "LAE: Language-aware encoder for monolingual and multilingual ASR." arXiv preprint arXiv:2206.02093, 2022.

**[11]** Ma, G, Wang, W, Li, Y, Yang, Y, Du, B, Fu, H. "LAE-ST-MOE: Boosted language-aware encoder using speech translation auxiliary task for E2E code-switching ASR." In 2023 IEEE Automatic Speech Recognition and Understanding Workshop (ASRU), pp. 1-8. IEEE, 2023.

**[12]** Yang, A, Nagrani, A, Seo, P H, Miech, A, Pont-Tuset, J, Laptev, I, Sivic, J, Schmid, C. "Vid2seq: Large-scale pretraining of a visual language model for dense video captioning." In Proceedings of the IEEE/CVF Conference on Computer Vision and Pattern Recognition, pp. 10714-10726. 2023.

**[13]** Buch, S, Eyzaguirre, C, Gaidon, A, Wu, J, Fei-Fei, L, Niebles, J C. "Revisiting the 'video' in video-language understanding." In Proceedings of the IEEE/CVF Conference on Computer Vision and Pattern Recognition, pp. 2917-2927. 2022.

**[14]** Sun, C, Myers, A, Vondrick, C, Murphy, K, Schmid, C. "VideoBERT: A joint model for video and language representation learning." In Proceedings of the IEEE/CVF International Conference on Computer Vision, pp. 7464-7473. 2019.

**[15]** Xu, Z, Zhang, Y, Xie, E, Zhao, Z, Guo, Y, Wong, K-Y K, Li, Z, Zhao, H. "DriveGPT4: Interpretable end-to-end autonomous driving via large language model." IEEE Robotics and Automation Letters, 2024.

**[16]** Xu, H, Ghosh, G, Huang, P-Y, Arora, P, Aminzadeh, M, Feichtenhofer, C, Metze, F, Zettlemoyer, L. "VLM: Task-agnostic video-language model pre-training for video understanding." In Findings of the Association for Computational Linguistics: ACL-IJCNLP 2021, pp. 4227-4239. 2021.

**[17]** Wang, M, Zhang, J, Mei, J, Liu, Y, Jiang, Y. "ActionCLIP: Adapting language-image pretrained models for video action recognition." IEEE Transactions on Neural Networks and Learning Systems, 2023.

**[18]** Wang, N, Zhu, G, Li, H S, Zhang, L, Shah, S A A, Bennamoun, M. "Language model guided interpretable video action reasoning." In Proceedings of the IEEE/CVF Conference on Computer Vision and Pattern Recognition, pp. 18878-18887. 2024.

**[19]** Luo, H, Ji, L, Zhong, M, Chen, Y, Lei, W, Duan, N, Li, T. "CLIP4CLIP: An empirical study of CLIP for end to end video clip retrieval and captioning." Neurocomputing, 2022, 508: 293-304.

**[20]** Sudhakaran, S, Escalera, S, Lanz, O. "Gate-shift networks for video action recognition." In Proceedings of the IEEE/CVF Conference on Computer Vision and Pattern Recognition, pp. 1102-1111. 2020.

**[21]** Li, Y, Ji, B, Shi, J, Zhang, J, Kang, B, Wang, L. "TEA: Temporal excitation and aggregation for action recognition." In Proceedings of the IEEE/CVF Conference on Computer Vision and Pattern Recognition, pp. 909-918. 2020.

**[22]** Wang, Z, She, Q, Smolic, A. "Action-net: Multipath excitation for action recognition." In Proceedings of the IEEE/CVF Conference on Computer Vision and Pattern Recognition, pp. 13214-13223. 2021.

**[23]** Ma, P, Martinez, B, Petridis, S, Pantic, M. "Towards practical lipreading with distilled and efficient models." In ICASSP 2021-2021 IEEE International Conference on Acoustics, Speech and Signal Processing (ICASSP), pp. 7608-7612. IEEE, 2021.

**[24]** Shrivastava, N, Saxena, A, Kumar, Y, Shah, R R, Mahata, D, Stent, A. "MobiVSR: A visual speech recognition solution for mobile devices." arXiv preprint arXiv:1905.03968, 2019.

**[25]** Koumparoulis, A, Potamianos, G, Thomas, S, da Silva Morais, E. "Resource-adaptive deep learning for visual speech recognition." In INTERSPEECH, pp. 3510-3514. 2020.

---

> ### Author Rebuttal · Authors · 2024-08-17
>
> **[26]** Haliassos, A, Vougioukas, K, Petridis, S, Pantic, M. "Lips don't lie: A generalisable and robust approach to face forgery detection." In Proceedings of the IEEE/CVF Conference on Computer Vision and Pattern Recognition, pp. 5039-5049. 2021.
>
> **[27]** Zhu, Z, Yang, H, Tang, M, Yang, Z, Eskimez, S E, Wang, H. "Real-time audio-visual end-to-end speech enhancement." In ICASSP 2023-2023 IEEE International Conference on Acoustics, Speech and Signal Processing (ICASSP), pp. 1-5. IEEE, 2023.
>
> **[28]** Koumparoulis, A, Potamianos, G. "Accurate and resource-efficient lipreading with EfficientNetV2 and transformers." In ICASSP 2022-2022 IEEE International Conference on Acoustics, Speech and Signal Processing (ICASSP), pp. 8467-8471. IEEE, 2022.
>
> **[29]** Gulati, A, Qin, J, Chiu, C-C, Parmar, N, Zhang, Y, Yu, J, Han, W, et al. "Conformer: Convolution-augmented transformer for speech recognition." arXiv preprint arXiv:2005.08100, 2020.
>
> **[30]** Lu, Y, Huang, M, Li, H, Guo, J, Qian, Y. "Bi-encoder transformer network for Mandarin-English code-switching speech recognition using mixture of experts." In INTERSPEECH, pp. 4766-4770. 2020.
>
> **[31]** Yan, B, Zhang, C, Yu, M, Zhang, S-F, Dalmia, S, Berrebbi, D, Weng, C, Watanabe, S, Yu, D. "Joint modeling of code-switched and monolingual ASR via conditional factorization." In ICASSP 2022-2022 IEEE International Conference on Acoustics, Speech and Signal Processing (ICASSP), pp. 6412-6416. IEEE, 2022.
>
> **[32]** Song, T, Xu, Q, Ge, M, Wang, L, Shi, H, Lv, Y, Lin, Y, Dang, J. "Language-specific characteristic assistance for code-switching speech recognition." 2022.
>
> **[33]** Song, T, Xu, Q, Lu, H, Wang, L, Shi, H, Lin, Y, Yang, Y, Dang, J. "Monolingual recognizers fusion for code-switching speech recognition." arXiv preprint arXiv:2211.01046, 2022.
>
> **[34]** Chen, P, Yu, F, Liang, Y, Xue, H, Wan, X, Zheng, N, Zhou, H, Xie, L. "BA-MoE: Boundary-aware mixture-of-experts adapter for code-switching speech recognition." In 2023 IEEE Automatic Speech Recognition and Understanding Workshop (ASRU), pp. 1-7. IEEE, 2023.
>
> **[35]** Tan, F, Feng, C, Wei, T, Gong, S, Leng, X, Chu, W, Ma, J, Wang, S, Gao, Y. "Improving end-to-end modeling for Mandarin-English code-switching using lightweight switch-routing mixture-of-experts." In Proc. INTERSPEECH, vol. 2023, pp. 4224-4228. 2023.
>
> **[36]** Wang, W, Ma, G, Li, Y, Du, B. "Language-routing mixture of experts for multilingual and code-switching speech recognition." arXiv preprint arXiv:2307.05956, 2023.
>
> **[37]** Deng, S, Li, C, Bai, F, Zhang, Q, Zhang, W-Q, Yang, R, Cheng, G, Zhang, P, Yan, Y. "Summary on the ISCSLP 2022 Chinese-English code-switching ASR challenge." In 2022 13th International Symposium on Chinese Spoken Language Processing (ISCSLP), pp. 527-531. IEEE, 2022.
>
> **[38]** Malik, M, Malik, M K, Mehmood, K, Makhdoom, I. "Automatic speech recognition: a survey." Multimedia Tools and Applications, 80 (2021): 9411-9457.
>
> **[39]** Sheng, C, Kuang, G, Bai, L, Hou, C, Guo, Y, Xu, X, Pietikäinen, M, Liu, L. "Deep learning for visual speech analysis: A survey." IEEE Transactions on Pattern Analysis and Machine Intelligence, 2024.
>
> **[40]** Luo, M, Yang, S, Chen, X, Liu, Z, Shan, S. "Synchronous Bidirectional Learning for Multilingual Lip Reading."
>
> **[41]** Chung, J S, Zisserman, A. "Lip reading in the wild." In Computer Vision–ACCV 2016: 13th Asian Conference on Computer Vision, Taipei, Taiwan, November 20-24, 2016, Revised Selected Papers, Part II 13, pp. 87-103. Springer International Publishing, 2017.
>
> **[42]** Yang, S, Zhang, Y, Feng, D, Yang, M, Wang, C, Rao, G, Long, K, Shan, S, Chen, X. "LRW-1000: A naturally-distributed large-scale benchmark for lip reading in the wild." In 2019 14th IEEE international conference on automatic face & gesture recognition (FG 2019), pp. 1-8. IEEE, 2019.
>
> **[43]** Son Chung, J, Senior, A, Vinyals, O, Zisserman, A. "Lip reading sentences in the wild." In Proceedings of the IEEE conference on computer vision and pattern recognition, pp. 6447-6456. 2017.
>
> **[44]** Tseng, L-H, Fu, Y-K, Chang, H-J, Lee, H-y. "Mandarin-English Code-switching Speech Recognition with Self-supervised Speech Representation Models." (2022).
>
> **[45]** Tan, Ganchao, et al. "Multi-grained spatio-temporal features perceived network for event-based lip-reading." Proceedings of the IEEE/CVF Conference on Computer Vision and Pattern Recognition. 2022.
>
> **[46]** Jo, Byungho, et al. "IFQA: interpretable face quality assessment." Proceedings of the IEEE/CVF winter conference on applications of computer vision. 2023.

---

### Decision · Program_Chairs · 2024-09-26

**Decision:**

Accept (Poster)

**Comment:**

I would like to congratulate the authors on their work and efforts, both before and after submission Overall, there was a strong consensus among the reviewers, all of whom recommended acceptance of the paper. In front of this, the following meta-review can simply recognize this consensus and aim to extract the key points raised by the reviewers regarding quality, originality, significance, and the list of major pros and cons. Of course, the following meta-review assumes that the commitments/promises made by the authors in the rebuttal have been applied, and that pending ones will be fulfilled before camera-ready submission.


## Quality

This work explicitly addressed the code-switching phenomenon (i.e., the introduction of words from a second language into a sentence spoken in a primary language) in lip reading scenarios. As this is a novel task, the authors contributed with a new dataset, appropriate benchmarking experiments, and – also worth mentioning – an open-source app for lip-reading data collection.

The dataset is competitively large, comparable to existing monolingual lip-reading datasets in terms of hours recorded and number of subjects (with a few exceptions). It comprises around 85k spoken sentences from a pool of 300 scripted sentences carefully scripted to ensure a balanced variety of words and topics. The dataset and annotations were manually curated to ensure high-quality samples and correct lip-reading transcriptions. Criticisms from o2C4 and Rsda were that the dataset focused on a single language pair (i.e. English-Chinese), had too few speakers (62), had limited ethnic diversity (Chinese only) and overly controlled videos (e.g. no clutter in the background or small viewpoint variations). As these limitations could not be addressed retrospectively, the authors could only acknowledge them and promise a future expansion with more people, ethnicities, and appearance variation. In addition, A5e4 pointed out the short duration of videos (3 seconds) and, hence, lack of context. However, this seems to be standard for other lip-reading datasets, leaving long-context lip-reading left as future work.

The baseline models selected for benchmarking were appropriate and well justified. The results were quite satisfactory in terms of the metrics evaluated (WER, CER, and MER), showing how this task can be accomplished while still leaving considerable room for improvement to encourage further research. Concerns about the size of the dataset and generalization were raised by the reviewers (8wJh), but the authors performed additional experiments, which did not reveal signs of overfitting/bias.
The lip-reading data collection app was made public (as clarified by the authors in the rebuttal), a potentially valuable resource for future researchers working on this topic.


## Clarity

The paper was already quite clear prior to revision, especially in terms of writing. The focus and structure were initially criticized but have since been improved. Upon request by Rsda to emphasize the dataset more than the baseline models, the text containing the models’ details was moved to supplementary material, allowing for the inclusion of additional details about the dataset composition.


## Originality

Prior to this work, code-switching lip-reading datasets/benchmarks did not exist in the literature. This is the first work that poses this as a problem and contributes with data and baselines to address it.
Note the proposed dataset is not a blend of other existing datasets, but that had to be collected from scratch, annotated, and curated. As discussed between the authors and reviewer Rsda, monolingual and/or multilingual datasets do not serve for the purpose of creating a code-switching lip-reading dataset.

## Significance
In a globalized world, where multi-cultural and multi-lingual communication is common, there is a strong need to address code-switching. Despite limitations, such as the focus on a single language pair or limited ethnic diversity, this work significantly broadens the applicability of lip-reading systems in real-world scenarios, very much necessary for individuals with hearing impairments.


## Pros and Cons

**Pros:**
- New task identified and addressed: code-switching lip-reading.
- New large-enough dataset to be released under CC-BY-4.0 license.
- Spoken sentences cover a wide range of topics and varying ratios of code-switching.
- The use of phone cameras for recordings enhances the real-world applicability.
- Solid baselines were benchmarked while leaving room for improvement.
- The work seems not raise ethical concerns or be very minor. Informed consent forms were used and video data to be released under npz format.
- Potential negative societal impacts were discussed (in response to Rsda) and included in the manuscript.

**Cons:**
- Only one ethnicity.
- Chinese and English are the two only languages.
- Controlled scenarios, limiting in-the-wild application.